# POLICY DECORATOR: MODEL-AGNOSTIC ONLINE REFINEMENT FOR LARGE POLICY MODEL

**Xiu Yuan**[∗], **Tongzhou Mu**[∗], **Stone Tao, Yunhao Fang, Mengke Zhang, Hao Su**
UC San Diego
{x1yuan, t3mu, stao, yuf026, mezhang, haosu}@ucsd.edu

## ABSTRACT

Recent advancements in robot learning have used imitation learning with large models and extensive demonstrations to develop effective policies. However, these models are often limited by the quantity, quality, and diversity of demonstrations. This paper explores improving offline-trained imitation learning models through online interactions with the environment. We introduce Policy Decorator, which uses a model-agnostic residual policy to refine large imitation learning models during online interactions. By implementing controlled exploration strategies, Policy Decorator enables stable, sample-efficient online learning. Our evaluation spans eight tasks across two benchmarks—ManiSkill and Adroit—and involves two state-of-the-art imitation learning models (Behavior Transformer and Diffusion Policy). The results show Policy Decorator effectively improves the offline-trained policies and preserves the smooth motion of imitation learning models, avoiding the erratic behaviors of pure RL policies. See our project page for videos.

## 1 INTRODUCTION

Encouraged by the recent success of large language and vision foundation models (Brown et al., 2020; Kirillov et al., 2023), the field of robot learning has seen significant advances through **imitation learning** (particularly behavior cloning), where large models leverage extensive robotic demonstrations to develop effective policies (Bousmalis et al., 2023; Brohan et al., 2022; 2023; Ahn et al., 2022). Despite these advancements, the performance of learned models is limited by the quantity, quality, and diversity of pre-collected demonstration data. This limitation often prevents models from handling all potential corner cases, as *demonstrations cannot cover every possible scenario (e.g., test-time objects can be entirely different from training ones)*. Unlike NLP and CV, scaling up demonstration collection in robotics, such as RT-1

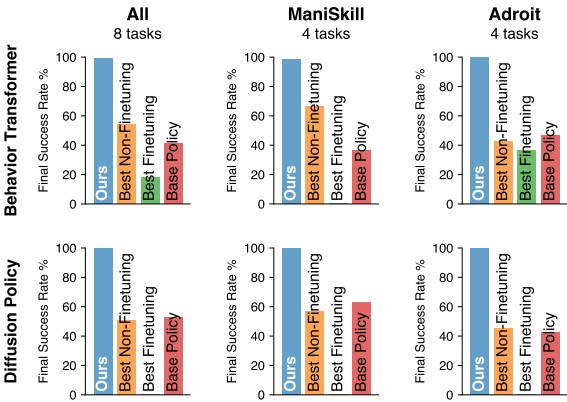

Figure 1: Policy Decorator improves base policy to near-perfect performance on two benchmarks, outperforming fine-tuning and non-fine-tuning baselines.

(Brohan et al., 2022) and Open X-Embodiment (Collaboration, 2023), requires extensive time and resources, involving years of data collection by numerous human teleoperators, making it costly and time-consuming. In contrast, cognitive research indicates that infants acquire skills through active interaction with their environment rather than merely observing others (Saylor & Ganea, 2018; Sheya & Smith, 2010; Adolph et al., 1997; Corbetta, 2021). This raises a natural question: *Can we further improve an offline-trained large policy through online interactions with the environment?*

The most straightforward approach to improving an offline-trained imitation learning policy is to fine-tune it using reinforcement learning (RL) with a sparse reward (Kumar et al., 2022; Yang et al., 2023a). However, several challenges hinder this strategy. *Firstly*, many state-of-the-art imitation

---

[∗]These authors contributed equally to this work.

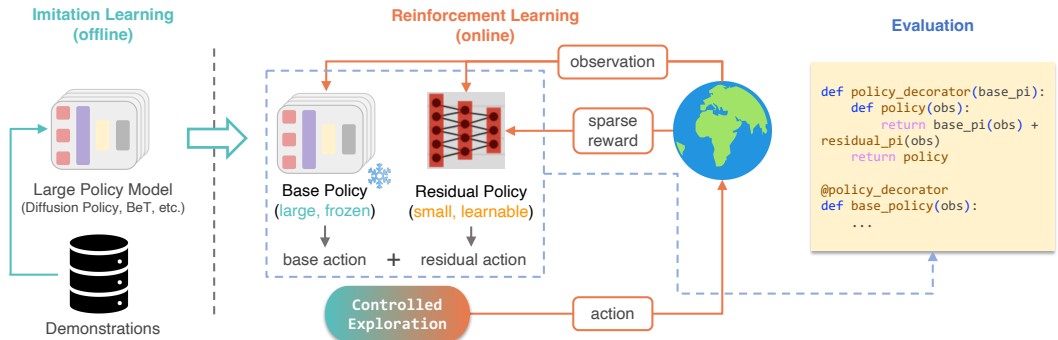

Figure 2: **Our framework (Policy Decorator)** improves large policy models through online interactions. We learn a residual policy via RL using controlled exploration strategies (Sec. 4.2). Once learned, it functions similarly to Python decorators—wrapping the base policy with an additional function to boost performance.

learning models have specific designs to accommodate the multimodal action distributions in the demonstrations, which unfortunately make them *non-trivial to fine-tune using RL*. For example, Behavior Transformer (Shafiullah et al., 2022), MCNN (Sridhar et al., 2023), and VINN (Pari et al., 2021) all incorporate some non-differentiable components (clustering, nearest neighbor search) which are incompatible with the gradient-based optimization in RL. Similarly, Diffusion Policy (Chi et al., 2023) requires ground truth action labels to supervise its denoising process, but these action labels are unavailable in RL setups (refer to Appendix H.1 for a more detailed discussion). *Secondly*, even if an imitation learning model were compatible with RL, the fine-tuning process would be prohibitively costly for two reasons: 1) the increasing number of parameters in modern large policy models, and 2) the extensive gradient updates required during sparse-reward RL training, a process known for its poor sample efficiency.

To devise a method for online improvement, we must first understand why an offline-trained imitation learning policy sometimes fails to solve tasks. As studied in (Ross et al., 2011; Ross & Bagnell, 2010; Syed & Schapire, 2010; Chang et al., 2015; Xu et al., 2020), a major issue with policies learned from purely offline data is **compounding error**. Small errors gradually accumulate, eventually leading the policy to states not covered in the demonstration dataset. However, correcting these errors may only require minimal effort. Even if the final trajectory deviates significantly from the correct path, slight adjustments can bring it back on track, as shown in Fig. 3. In other words, the model only needs "refinement" for the finer parts of the tasks. Modeling such small adjustments typically does not necessitate complex architectures or large numbers of parameters. Therefore, we propose to online learn a **residual policy** (parameterized by a small network) to correct the behavior of the offline-trained imitation learning models, referred to as the "base policy" throughout this paper. This approach addresses the incompatibility between models and RL and avoids the costly gradient updates on large models.

While learning a residual policy through online RL (Jo-hannink et al., 2019; Alakuijala et al., 2021; Silver et al., 2018; Zhang et al., 2019) can, in principle, refine a base policy, practical implementation is still challenging. As demonstrated in our experiments (Sec. 5), without constraints, random exploration during RL training often leads to failure in tasks requiring precise control, resulting in no learning signals in sparse reward settings (see this video

Figure 3: Small adjustments can bring deviated trajectories back on track.

for an example). To overcome these challenges and enable stable, sample-efficient learning, we propose a set of strategies to ensure the RL agent (with the residual policy) **explores the environment in a controlled manner**. This approach ensures that the agent continuously receives sufficient success signals while adequately exploring the environment. We call this framework the **Policy Decorator** because it functions similarly to decorators in Python—enhancing the original policy by wrapping it with an additional function to boost its performance, as illustrated in Fig. 2. Like Python decorators, our framework does not require any prior knowledge of the original policy and treats it as a black box, making it **model-agnostic**.

We evaluate our approach across a variety of benchmarks and imitation learning models. Specifically, we examine 8 tasks from 2 benchmarks: ManiSkill (Mu et al., 2021; Gu et al., 2023) and Adroit

(Rajeswaran et al., 2017), in conjunction with 2 state-of-the-art imitation learning models: Behavior Transformer (Shafiullah et al., 2022) (action clustering + regression) and Diffusion Policy (Chi et al., 2023) (diffusion models + receding horizon control). Our results demonstrate that the Policy Decorator consistently improves various offline-trained large policy models to near-optimal performance in most cases. Furthermore, the learned policy maintains the desirable properties of the imitation learning policy, producing smooth motions rather than the jerky motions generated by pure RL policies.

To summarize, our contributions are as follows:

- Conceptually, we raise the critical research question: "How can large policy models be improved through online interactions?", and identify limitations of fine-tuning and vanilla residual RL.
- Technically, we propose Policy Decorator, a *model-agnostic* framework for refining large policy models through online environmental interactions.
- Empirically, we conduct extensive experiments on 8 challenging robotic tasks and 2 state-of-the-art imitation learning models, demonstrating Policy Decorator's advantages in both task performance and learned policy properties.

## 2    RELATED WORKS

**Learning from Demo**  Learning control policies through trial and error can be inefficient and unstable, prompting research into leveraging demonstrations to enhance online learning. Demonstrations can be utilized through pure offline imitation learning, including behavior cloning (Pomerleau, 1988) and inverse reinforcement learning (Ng et al., 2000). Alternatively, demonstrations can be incorporated during online learning, serving as off-policy experience (Mnih et al., 2015; Hessel et al., 2018; Ball et al., 2023; Nair et al., 2018) or for on-policy regularization (Kang et al., 2018; Rajeswaran et al., 2017). Furthermore, demonstrations can be used to estimate reward functions for RL problems (Xie et al., 2018; Aytar et al., 2018; Vecerik et al., 2019; Zolna et al., 2020; Singh et al., 2019). When the offline dataset includes both demonstrations and negative trajectories, offline-to-online RL approaches first apply offline RL to learn effective policy and value initializations from offline data, followed by online fine-tuning (Nair et al., 2020; Kostrikov et al., 2021; Lyu et al., 2022; Nakamoto et al., 2024). *In this work, we adopt a more direct approach to utilize demonstrations: distilling demonstrations into a large policy model and subsequently improving it through online interactions.*

**Residual Learning**  The concept of learning residual components has been widely applied across various domains, including addressing the vanishing gradient problem in deep neural networks (He et al., 2016; Vaswani, 2017) and parameter-efficient fine-tuning (Hu et al., 2021). In robotic control, researchers have employed online RL to learn corrective residual components for various base policies, such as hand-crafted controllers (Johannink et al., 2019), non-parametric models (Haldar et al., 2023b), and pre-trained neural networks (Alakuijala et al., 2021; Ankile et al., 2024). Residual learning can also be achieved through supervised learning (Jiang et al., 2024). Our work focuses on the online improvement of *large policy models*, identifying residual policy learning as an ideal solution due to its model-agnostic nature. We highlight the *uncontrolled exploration issue in vanilla residual RL*, propose a set of strategies to address it, and further enhance its efficiency through careful examination of design choices.

**Advanced Imitation Learning**  Imitation learning provides an effective approach to teaching robots complex skills. However, naive imitation learning often struggles to model multi-modal distributions within demonstration datasets (Chi et al., 2023). Several advanced methods have been proposed to address this limitation: Shafiullah et al. (2022); Lee et al. (2024); Sridhar et al. (2023); Pari et al. (2021) represent actions as discrete values with offsets, Florence et al. (2022) employs energy-based models, and Chi et al. (2023) leverages diffusion models. While these techniques effectively learn from multi-modal data, they often create models that are *non-trivial to fine-tune using RL*. Even if they were compatible with RL, *the fine-tuning process can be computationally prohibitive* due to the large number of parameters in modern policy models. These limitations motivate our approach of using online residual policy learning to improve imitation learning models.

## 3    PROBLEM SETUP

In this paper, we focus on improving an offline-trained large policy (referred to as "base policy") through online interactions. We make the following assumptions:

1. An environment is available for online interactions with task success signals (sparse rewards).

2. The base policy may have a large number of parameters or complex architectures, making fine-tuning non-trivial or computationally expensive. This assumption holds for many modern large policy models (Brohan et al., 2022; 2023; Chi et al., 2023; Shafiullah et al., 2022).

3. The base policy exhibits reasonable initial performance, though not perfect (i.e., it can make progress towards task completion, which is achievable by many state-of-the-art IL methods with a reasonable amount of demonstrations). An excessively poor base policy is not worth improving.

Note that our approach does not make any specific assumptions about model architectures or training methods. Instead, we treat these models as *black boxes* that take observations as input and produce actions as output. In our experiments, we choose to improve base policies trained by imitation learning rather than offline RL policies. This is because: 1) collecting demonstrations alone is more cost-effective and thus more common; 2) as demonstrated in multiple studies (Mandlekar et al., 2021; Florence et al., 2022), imitation learning outperforms offline RL in demonstration-only settings.

## 4 POLICY DECORATOR: MODEL-AGNOSTIC ONLINE REFINEMENT

In this work, our goal is to online improve a large policy model, which is usually offline-trained by imitation learning and usually has some specific designs in model architecture. To this end, we propose *Policy Decorator*, a model-agnostic framework for refining large policy models via online interactions with environments. Fig. 2 provides an overview of our framework.

Policy Decorator is grounded on **learning a residual policy via reinforcement learning with sparse rewards**, which is described in Sec. 4.1. On top of it, we devise a set of strategies to ensure the RL agent (in combination with the base policy and the residual policy) explores the environment in a controlled manner. Such a **controlled exploration** mechanism is detailed in Sec. 4.2. Finally, we discuss several important **design choices** that further enhance learning efficiency in Sec. 4.3.

### 4.1 LEARNING RESIDUAL POLICY VIA RL

Given the base policy $\pi_{base}$, we then train a residual policy $\pi_{res}$ on top of it using reinforcement learning. The base policy $\pi_{base}$ can be either deterministic (e.g., Behavior Transformer (Shafiullah et al., 2022)) or stochastic (e.g., Diffusion Policy (Chi et al., 2023)), and it remains frozen during the RL process. The residual policy $\pi_{res}$ is updated through RL gradients, so it should be a differentiable function compatible with RL gradients. In this work, we model the residual policy $\pi_{res}$ as a Gaussian policy parameterized by a small neural network (either an MLP or a CNN, depending on the observation modality). To interact with the environment, the actions from both policies are combined by summing their output actions, i.e., the action executed in the environment is $\pi_{base}(s) + \pi_{res}(s)$. For stochastic policies, actions are sampled individually from both policies and then summed.

The residual policy is trained to maximize the expected discounted return derived from the sparse reward (i.e., the task's success signal). We employ the Soft Actor-Critic (SAC) algorithm (Haarnoja et al., 2018) due to its superior sample efficiency and stability. Several important design choices arise when implementing SAC for learning the residual policy, which we discuss in Sec. 4.3. Our method is also compatible with PPO (Schulman et al., 2017), as illustrated in Appendix D.3.

### 4.2 CONTROLLED EXPLORATION

While learning a residual policy by RL can in principle refine a base policy, practical implementation can be challenging. As demonstrated in our experiments (Sec. 5), without constraints, random exploration during RL training often leads to failure in tasks requiring precise control, resulting in no learning signals in sparse reward settings (see this video for an example). To overcome these challenges and enable stable, sample-efficient learning, we propose a set of strategies ensuring the RL agent (in combination with the base policy and the residual policy) **explores the environment in a controlled manner**. The goal is to make sure the agent continuously receives sufficient success signals while adequately exploring the environment.

**Bounded Residual Action** When using the residual policy to correct the base policy, we do not want the resulting trajectory to deviate too much from the original trajectory because it usually leads to failure. Instead, we expect the residual policy to only make a bit "refinement" at the finer parts of the

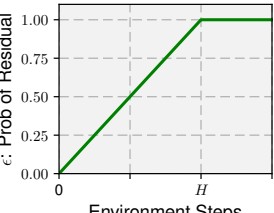

Figure 5: **Tasks Visualizations.** ManiSkill (left four figures) and Adroit (right four figures).

tasks. To reflect this spirit, we bound the output of the residual policy within a certain scale. Since we use SAC as our backbone RL algorithm, the output of the policy is naturally bounded by a squashing function (tanh), whose range is $(-1, 1)$. We further scale the action sampled from the Gaussian policy with a hyperparameter $\alpha$, making the range of the residual action $(-\alpha, \alpha)$. We found that an appropriate scale of residual action bound can be crucial for some precise tasks. We investigated the effects of hyperparameter $\alpha$ in Sec. 5.4.2.

**Progressive Exploration Schedule** Given that our residual policy is randomly initialized, the agent (combined with the base policy and residual policy) may exhibit highly random behavior and fail to succeed at the initial stage of learning. Therefore, the base policy alone, trained by imitation learning, can be more reliable during the early stages. As training progresses, the residual policy can be gradually improved, making it safer to incorporate its suggestions.

Inspired by the $\epsilon$-greedy strategy used in DQN (Mnih et al., 2015), we propose to progressively introduce actions from the residual policy into the agent's behavior policy. Specifically, the behavior policy will use actions from the residual policy to complement the base policy with probability $\epsilon$ and rely solely on the base policy with probability $1 - \epsilon$. Formally, during training,

$$\pi_{behavior}(s) = \begin{cases} \pi_{base}(s) + \pi_{res}(s) & \text{Uniform}(0, 1) < \epsilon \\ \pi_{base}(s) & \text{otherwise} \end{cases} \quad (1)$$

Figure 4: Progressive Exploration Schedule.

The parameter $\epsilon$ increases linearly from 0 to 1 over a specified number of time steps, as shown in Fig. 4, where $H$ is a hyperparameter. Our experiments in Sec. 5.4.2 indicate that while tuning $H$ can enhance sample efficiency, using a large $H$ is generally a safe choice.

### 4.3 DESIGN CHOICES & IMPLEMENTATION DETAILS

We investigated a few important design choices in the implementation of Policy Decorator, with supporting experiments provided in Appendix E.

**Input of Residual Policy** The residual policy can receive input in the form of either observation alone or both observation and action from the base policy. Our experiments indicate that using only observation typically produces better results, as illustrated in Fig. 19.

**Input of Critic** In SAC, the critic $Q(s, a)$ takes an action as input, and there are several design choices regarding this action: we can use 1) the sum of the base action and residual action; 2) the concatenation of both; or 3) the residual action alone. Based on our experiments shown in Fig. 20, using the sum of both actions yields the best performance.

## 5 EXPERIMENTS

The goal of our experimental evaluation is to study the following questions:

1. Can Policy Decorator **effectively refine offline-trained imitation policies using online RL with sparse rewards** under different setups (different tasks, base policy architectures, demonstration sources, and observation modalities)? (Sec. 5.3)

2. What are the **effects of the components** introduced by the Policy Decorator? (Sec. 5.4)

3. Does Policy Decorator generate **better task-solving behaviors** compared to other types of learning paradigms (e.g., pure IL and pure RL)? (Sec. 5.5)

## 5.1 EXPERIMENTAL SETUP

To validate Policy Decorator's versatility, our experimental setup incorporates *variations across the following dimensions*:

- **Task Types:** Stationary robot arm manipulation, mobile manipulation, dual-arm coordination, dexterous hand manipulation, articulated object manipulation, and high-precision tasks. Fig. 5 illustrates sample tasks from each benchmark.
- **Base Policies:** Behavior Transformer and Diffusion Policy.
- **Demo Sources:** Teleoperation, Task and Motion Planning, RL, and Model Predictive Control.
- **Observation Modalities:** State observation (low-dim) and visual observation (high-dim).

We summarize the key details of our setups as follows (further details on the *task descriptions, demonstrations, and base policy implementation* can be found in Appendix A and B.1).

### 5.1.1 TASK DESCRIPTION

Our experiments are conducted on 8 tasks across 2 benchmarks: ManiSkill (robotic manipulation; 4 tasks), and Adroit (dexterous manipulation; 4 tasks). See Fig. 5 for illustrations.

**ManiSkill** We consider four challenging tasks from ManiSkill. StackCube and PegInsertionSide demand *high-precision control*, with PegInsertion featuring a mere 3mm clearance. TurnFaucet and PushChair introduce *object variations*, where the base policy is trained on source environment objects, but target environments for online interactions contain different objects. These complexities make it **challenging for pure offline imitation learning to achieve near-perfect success rates**, necessitating online learning approaches. For all ManiSkill tasks, we use 1000 demonstrations provided by the benchmark (Mu et al., 2021; Gu et al., 2023) across all methods. These demonstrations are generated through task and motion planning, model predictive control, and reinforcement learning.

**Adroit** We consider all four dexterous manipulation tasks from Adroit: Door, Hammer, Pen, and Relocate. The tasks should be solved using a complex, 24-DoF manipulator, simulating a real hand. For all Adroit tasks, we use 25 demonstrations provided by the original paper (Rajeswaran et al., 2017) for all methods. These demonstrations are collected by human teleoperation.

### 5.1.2 BASE POLICY MODEL

We selected two popular imitation learning models as our base policy models for improvement.

**Behavior Transformer** (Shafiullah et al., 2022) is a GPT-based policy architecture for behavior cloning. It handles multi-modal action distribution by representing an action as a combination of a cluster center (predicted by a classification head) and an offset (predicted by a regression head). The action cluster centers are determined by k means, which is non-differentiable, thus only the offset can be fine-tuned using RL gradients.

**Diffusion Policy** (Chi et al., 2023) is a state-of-the-art imitation learning method that leverages recent advancements in denoising diffusion probabilistic models. It generates robot action sequences through a conditional denoising diffusion process and employs action sequences with receding horizon control. The training of Diffusion Policy requires ground truth action labels to supervise its denoising process; however, these action labels are unavailable in RL setups, making the original training recipe incompatible with RL. Nevertheless, recent approaches have been developed to fine-tune diffusion models using RL in certain scenarios. See Appendix H for a more detailed discussion.

The implementation details of these two base policies can be found in Appendix B.1. To further demonstrate the versatility of our method, we also present the results on other types of base policies (including MLP, RNN, and CNN) in Appendix D.1.

## 5.2 BASELINES

We compare our approach against a set of strong baselines for online policy improvement, including both **fine-tuning-based methods** and **methods that do not involve fine-tuning**. A brief description of each baseline is provided below, with further implementation details available in Appendix B.5.

### 5.2.1 FINE-TUNING METHODS

As discussed in Sec. 1, making our base policies compatible with online RL is non-trivial. *We implemented several specific modifications to the base policies to enable fine-tuning*, as detailed in Appendix B.4. Since we consider the problem of improving large policy models where full-parameter fine-tuning can be costly, we employ LoRA (Hu et al., 2021) for parameter-efficient fine-tuning.

Our fine-tuning baseline selection follows this rationale: we first choose a basic RL algorithm for each base policy based on their specific properties, which serves as a basic baseline. Additionally, assuming access to the demonstrations used to train the base policies, we consider various learning-from-demonstration methods as potential baselines. Table 1 lists the most relevant learning-from-demo baselines. From these, we select *the strongest and most representative methods in each category* and implement them on top of the basic RL algorithm we initially selected.

**Basic RL** We use SAC (Haarnoja et al., 2018) as our basic fine-tuning method for Behavior Transformer, and use DIPO (Yang et al., 2023b) for Diffusion Policy (see Appendix H.2 for a discussion on other RL methods for Diffusion Policy). For both methods, we initialize the actor with the pre-trained base policy and use a randomly initialized MLP for the critic (refer to Appendix F.5.1 for an ablation study on this design choice). We also noticed a concurrent work (DPPO (Ren et al., 2024)) designed for fine-tuning Diffusion Policy using RL, and we compare our approach with it *on their tasks*. See Appendix C.1 for more details.

**RLPD** (Ball et al., 2023) is a state-of-the-art online learning-from-demonstration method that *utilizes demonstrations as off-policy experience*. It enhances vanilla SACfd with critic layer normalization, symmetric sampling, and sample-efficient RL techniques.

**ROT** (Haldar et al., 2023a) is a representative online learning-from-demonstration algorithm that *utilizes demonstrations to derive dense rewards* and *for policy regularization*. It adaptively combines offline behavior cloning with online trajectory-matching based rewards.

Table 1: **Potential Fine-tuning Baselines with Demos.** We categorize potential learn-from-demo baselines into four distinct categories, and choose the best and most representative methods from each category as our main points of comparison. Selected baselines are in **bold**.

| Category | Method |
|---|---|
| **Demo for Reward Learning** | **ROT** (Haldar et al., 2023a) |
| | GAIL (Ho & Ermon, 2016) |
| | DAC (Kostrikov et al., 2018) |
| **Demo as Off-Policy Experience** | **RLPD** (Ball et al., 2023) |
| | SACfd (Vecerik et al., 2017) |
| **Demo as On-Policy Regularization** | **ROT** (Haldar et al., 2023a) |
| | DAPG (Rajeswaran et al., 2017) |
| | AWAC (Nair et al., 2020) |
| **Offline RL Online Fine-tuning** | **Cal-QL** (Nakamoto et al., 2024) |
| | IQL (Kostrikov et al., 2021) |
| | CQL (Kumar et al., 2020) |

**Cal-QL** (Nakamoto et al., 2024) is a state-of-the-art *offline-to-online RL* method that "calibrates" the Q function in CQL (Kumar et al., 2020) for efficient online fine-tuning. In our setting, we use the same demonstration set used in other baselines as the offline data for Cal-QL. Unlike other fine-tuning baselines that initialize the critic randomly, Cal-QL can *potentially benefit from the pre-trained critic*.

### 5.2.2 NON-FINE-TUNING METHODS

**JSRL** (Uchendu et al., 2023) is a curriculum learning method that employs a guiding policy to bring the agent closer to the goal. In our setting, the pre-trained base policy serves as the guiding policy.

**Residual RL** (Johannink et al., 2019) learns a residual control signal on top of a hand-crafted conventional controller. Unlike our approach, *it explores the environment in an entirely uncontrolled manner*. For a fair comparison, we replace its hand-crafted controller with our base policies.

**FISH** (Haldar et al., 2023b) builds upon Residual RL by incorporating a non-parametric VINN (Pari et al., 2021) policy and learning an online offset actor with optimal transport rewards.

### 5.3 MAIN RESULTS & ANALYSIS

**Our Approach** We evaluate Policy Decorator with Behavior Transformer and Diffusion Policy as base policies, and the results are summarized in Fig. 6 and 7, respectively (see Fig. 1 for a barplot).

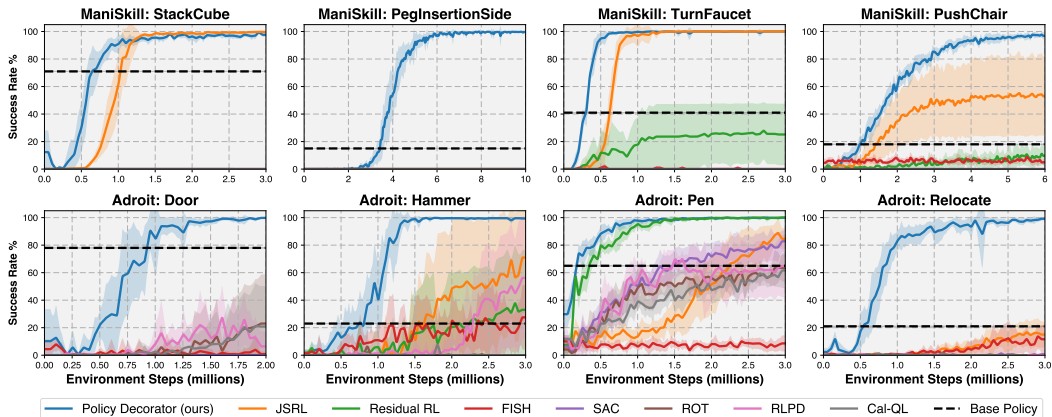

Figure 6: **Results (with Behavior Transformer):** During training, we evaluate the agent for 50 episodes every 50K environment steps. The curves depict the evaluation success rates averaged over ten seeds, and the shaded areas represent standard deviations. Our method consistently improves the base policy and outperforms all other baselines.

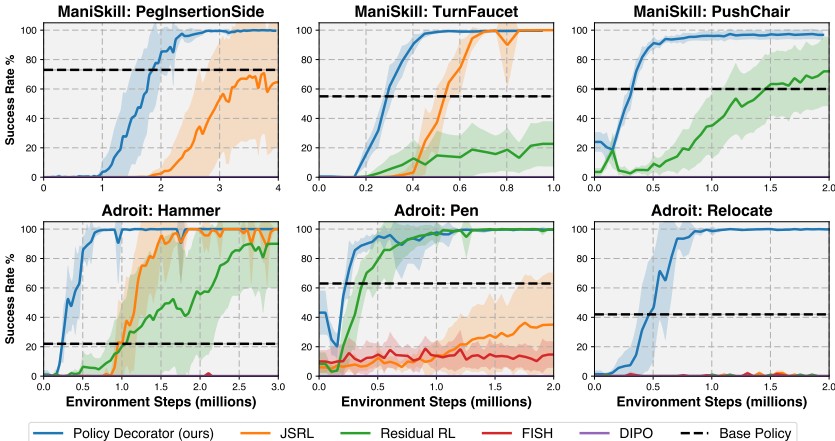

Figure 7: **Results (with Diffusion Policy):** The setup is similar to Fig. 6, but with different baselines due to the nature of Diffusion Policy. Since DIPO does not work at all on any tasks, we did not include other fine-tuning-based baselines built on top of DIPO. In addition, we did not test on StackCube and Adroit Door because the base policy is already near-optimal (99%+ success rates).

Policy Decorator improves the performance of both offline-trained policies to a near-perfect level on all tasks across ManiSkill and Adroit when given low-dimensional state observations. For Diffusion Policy, we did not test on StackCube and Door since the base policy already achieves near-optimal performance in these tasks.

**Non-Finetuning Baselines** Overall, JSRL performs the best among all baselines but only exceeds the base policy's performance on around half of the scenarios. Additionally, JSRL does not actually "improve" the base policy but instead learns an entirely new policy. This means that even if it achieves a high success rate, it does not preserve the desired properties of the original base policy, such as smooth and natural motion. See here for videos comparing the behavior of JSRL and the learned policy from our framework. Residual RL improves the base policy on 3 out of 6 tasks when combined with Diffusion Policy, but performs quite poorly when combined with Behavior Transformer. We suspect that this is because residual RL agents have a higher chance of obtaining task success signals through random exploration due to the stronger performance and robustness of the Diffusion Policy models. FISH performs poorly on most tasks, primarily due to the weak performance of the VINN. See Appendix G for detailed discussion on the failure of non-fine-tuning baselines.

**Finetuning Baselines** All fine-tuning-based baselines generally perform poorly in our evaluation. Cal-QL and RLPD can improve Behavior Transformer on a few Adroit tasks but completely fail on ManiSkill tasks. We suspect this is because *the randomly initialized critic network cannot*

Figure 8: **Results on Image Observations (with Diffusion Policy):** Similar to Fig. 7, but using image observations instead of low-dimensional state observations. We selected two tasks with complex visual appearances from each benchmark.

*provide meaningful gradients* and quickly causes the agent to deviate significantly from the original trajectories. In contrast, our controlled exploration strategies help the agent remain exposed to success signals. While Cal-QL can theoretically learn a good critic from offline data, we found that the learned critic does not aid online fine-tuning when it is trained purely on demonstration data without negative trajectories. This degradation over the course of Cal-QL online training has also been observed by Yang et al. (2023a). Another reason for the failure of fine-tuning-based methods is *the long-horizon nature of our tasks*. We observed that RL fine-tuning becomes effective when the task horizon is reduced (see Fig. 23). **A detailed analysis of the failure of fine-tuning baselines is presented in Appendix F.** For Diffusion Policy, we observed that DIPO failed to obtain any success signals across all tasks, so we did not further test other fine-tuning-based methods that rely on DIPO as the backbone RL algorithm. We hypothesize that this failure is due to the receding horizon control in Diffusion Policy, which complicates the fine-tuning process. For instance, when Diffusion Policy predicts 16 actions but only the first 8 are executed in the environment, *there is no clear method to supervise the latter 8 actions during fine-tuning*. Keeping the latter 8 actions unchanged is incorrect because once the first 8 actions are modified through fine-tuning, they may bring the agent to a new state where the latter 8 actions no longer apply.

**Visual Observations** Finally, we conducted experiments with visual observations. As shown in Fig. 8, the results validated that Policy Decorator also performs well with high-dim visual observations.

## 5.4 ABLATION STUDY

We conducted various ablations on Stack Cube and Push Chair tasks to provide further insights.

### 5.4.1 RELATIVE IMPORTANCE OF EACH COMPONENT

We examined the relative importance of Policy Decorator's main components: 1) residual policy learning; 2) progressive exploration schedule; and 3) bounded residual action. We thoroughly evaluated *all possible combinations* of these components, with results shown in Fig. 9. Each component greatly contributes to the overall performance, both individually and collectively. While residual policy learning establishes the foundation of our framework, using it alone does not sufficiently improve the base policy. Bounded residual action is essential for effective residual policy learning, and the progressive exploration schedule further enhances sample efficiency.

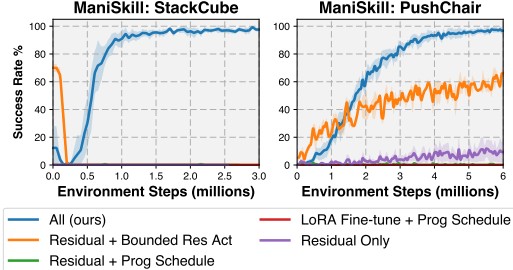

Figure 9: The importance of each component.

### 5.4.2 INFLUENCE OF KEY HYPERPARAMETERS

**Bound $\alpha$ of Residual Actions** The hyperparameter $\alpha$ determines the maximum adjustment the residual policy can make. Fig. 10 illustrates how $\alpha$ affects the learning process. If $\alpha$ is too small, the final performance may be adversely affected. Conversely, if $\alpha$ is too large, it may lead to poor sample efficiency during training. Although certain values achieve optimal sample efficiency, $\alpha$ values within a broad range (e.g., 0.1 to 0.5 for PushChair and 0.03 to 0.1 for StackCube) eventually converge to similar success rates, albeit with varying sample efficiencies. This indicates that **while**

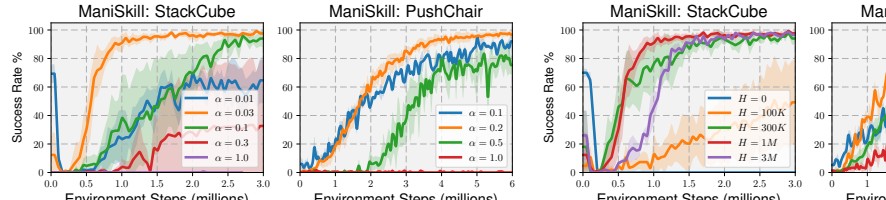

Figure 10: Different values of the bound $\alpha$ for Residual Actions.

Figure 11: Different values of $H$ in Progressive Exploration Schedule.

**the choice of $\alpha$ is impactful, our method remains robust across a wide range of $\alpha$ values**. In practice, tuning $\alpha$ is relatively straightforward: **we typically set it close to the action scale observed in the demonstration dataset** and make minor adjustments as necessary.

$H$ **in Progressive Exploration Schedule** The hyperparameter $H$ (see Fig. 4 for an illustration) controls the rate at which we switch from the base policy to the residual policy. From Fig. 11, we observe that a too-small $H$ can lead to complete failure due to aggressive exploration, while a large $H$ may result in relatively poor sample efficiency. Therefore, **tuning $H$ can enhance sample efficiency** and ensure stable training. However, **using a large $H$ is generally a safe choice** if sample efficiency is not the primary concern.

### 5.4.3 ADDITIONAL ABLATION STUDIES

Additional ablation studies are provided in Appendix D, with key conclusions summarized as follows:

- Policy Decorator also works with other types of base policies (e.g., MLP, RNN, and CNN). D.1
- Policy Decorator remains effective when applied to low-performing checkpoints. D.2
- Policy Decorator is also effective when using PPO as the backbone RL algorithm. D.3

### 5.5 PROPERTIES OF THE REFINED POLICY

An intriguing aspect of Policy Decorator is its ability to **combine the strengths of both Imitation Learning and Reinforcement Learning policies**. Previous observations have highlighted that robotic policies trained solely by RL often exhibit jerky actions, rendering them unsuitable for real-world application (Qin et al., 2022). Conversely, policies derived from demonstrations, whether from human teleoperation or motion planning, tend to produce more natural and smooth motions. However, the performance of such policies is constrained by the diversity and quantity of the demonstrations.

Our refined policy, learned through Policy Decorator, **achieves remarkably high success rates while retaining the favorable attributes of the base policy**. This is intuitive – by constraining residual actions, the resulting trajectory maintains proximity to the original trajectory, minimizing deviation.

Comparison with RL policies reveals that our refined approach exhibits significantly smoother behavior (see videos here). Furthermore, when compared with offline-trained base policies, our refined policy shows superior performance, effortlessly navigating through the finest part of the task (shown in this video), while preserving its multi-modal property (see Appendix J for details).

## 6 CONCLUSIONS, DISCUSSIONS, & LIMITATIONS

We propose the Policy Decorator framework, a flexible method for improving large behavior models using online interactions. We introduce controlled exploration strategies that boost the base policy's performance efficiently. Our method achieves near-perfect success rates on most tasks while preserving the smooth motions typically seen in imitation learning models, unlike the jerky movements often found in reinforcement learning policies.

**Limitations** Enhancing large models with online interactions requires significant training time and resources. While learning a small residual policy reduces computational costs compared to fully fine-tuning the large model, the process remains resource-intensive, especially for slow-inference models like diffusion policies. We found that only a few critical states need adjustment. Future research could focus on identifying and correcting these points more precisely to improve efficiency.

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

## A    FURTHER DETAILS ON THE EXPERIMENTAL SETUP

### A.1    TASK DESCRIPTIONS

We consider a total of 8 continuous control tasks from 2 benchmarks: ManiSkill (Mu et al., 2021), and Adroit (Rajeswaran et al., 2017). This section provides detailed task descriptions on overall information, task difficulty, object sets, state space, and action space. Some task details are listed in Table 2.

#### A.1.1    MANISKILL TASKS

For all tasks we evaluated on ManiSkill benchmark, we use consistent setup for state space, and action space. The state spaces adhere to a standardized template that includes proprioceptive robot state information, such as joint angles and velocities of the robot arm, and, if applicable, the mobile base. Additionally, task-specific goal information is included within the state. ManiSkill tasks we evaluated are very challenging because two of them require precise control and another two involve object variations. Below, we present the key details pertaining to the tasks used in this paper.

**Stack Cube**

- Overall Description: Pick up a red cube and place it onto a green one.
- Task Difficulty: This task requires precise control. The gripper needs to firmly grasp the red cube and accurately place it onto the green one.
- Object Variations: No object variations.
- Action Space: Delta position of the end-effector and joint positions of the gripper.
- State Observation Space: Proprioceptive robot state information, such as joint angles and velocities of the robot arm, and task-specific goal information.
- Visual Observation Space: one 64x64 RGBD image from a base camera and one 64x64 RGBD image from a hand camera.

**Peg Insertion Side**

- Overall Description: Insert a peg into the horizontal hole in a box.
- Task Difficulty: This task requires precise control. The gripper needs to firmly grasp the peg, perfectly align it horizontally to the hole, and inserts it.
- Object Variations: The box geometry is randomly generated.
- Action Space: Delta pose of the end-effector and joint positions of the gripper.
- State Observation Space: Proprioceptive robot state information, such as joint angles and velocities of the robot arm, and task-specific goal information.
- Visual Observation Space: one 64x64 RGBD image from a base camera and one 64x64 RGBD image from a hand camera.

**Turn Faucet**

- Overall Description: Turn on a faucet by rotating its handle.
- Task Difficulty: This task needs to handle object variations. The dataset contains trajectories of 10 faucet types, while in online interactions, the agent needs to deal with 4 novel faucets not present in the dataset. See Fig. 12.
- Object Variations: We have a source environment containing 10 faucets, and the dataset is collected in the source environment. The agent interacts with the target environment online, which contains 4 novel faucets.
- Action Space: Delta pose of the end-effector and joint positions of the gripper.
- State Observation Space: Proprioceptive robot state information, such as joint angles and velocities of the robot arm, the mobile base, and task-specific goal information.

- Visual Observation Space: one 64x64 RGBD image from a base camera and one 64x64 RGBD image from a hand camera.

**Push Chair**

- Overall Description: A dual-arm mobile robot needs to push a swivel chair to a target location on the ground (indicated by a red hemisphere) and prevent it from falling over. The friction and damping parameters for the chair joints are randomized.
- Task Difficulty: This task needs to handle object variations. The dataset contains trajectories of 5 chair types, while in online interactions, the agent needs to deal with 3 novel chairs not present in the dataset. See Fig. 12.
- Object Variations: We have a source environment containing 5 chairs, and the dataset is collected in the source environment. The agent interacts with the target environment online, which contains 3 novel chairs.
- Action Space: Joint velocities of the robot arm joints and mobile robot base, and joint positions of the gripper.
- State Observation Space: Proprioceptive robot state information, such as joint angles and velocities of the robot arm, task-specific goal information.
- Visual Observation Space: three 50x125 RGBD images from three cameras $120°$ apart from each other mounted on the robot.

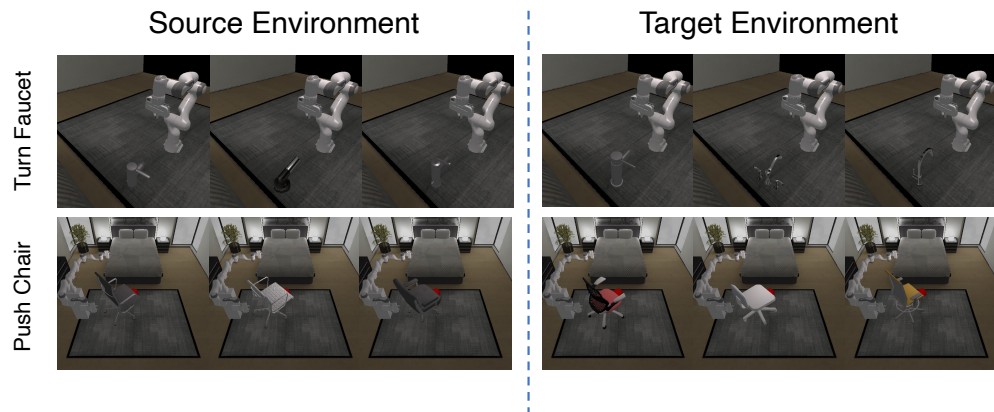

Figure 12: For the Turn Faucet and Push Chair tasks in the ManiSkill benchmark, *we have a source environment with various object instances from which the dataset is collected. The agent interacts with another target environment that includes unseen object instances.* This figure is for illustration purposes only. Please refer to the information above for specific details.

### A.1.2 ADROIT TASKS

**Adroit Door**

- Overall Description: The environment is based on the Adroit manipulation platform, a 28 degree of freedom system which consists of a 24 degrees of freedom ShadowHand and a 4 degree of freedom arm. The task to be completed consists on undoing the latch and swing the door open.
- Task Difficulty: The latch has significant dry friction and a bias torque that forces the door to stay closed. No information about the latch is explicitly provided. The position of the door is randomized.
- Object Variations: No object variations.
- Action Space: Absolute angular positions of the Adroit hand joints.
- State Observation Space: The angular position of the finger joints, the pose of the palm of the hand, as well as state of the latch and door.

- Visual Observation Space: one 128x128 RGB image from a third-person view camera.

**Adroit Pen**

- Overall Description: The environment is based on the Adroit manipulation platform, a 28 degree of freedom system which consists of a 24 degrees of freedom ShadowHand and a 4 degree of freedom arm. The task to be completed consists on repositioning the blue pen to match the orientation of the green target.
- Task Difficulty: The target is also randomized to cover all configurations.
- Object Variations: No object variations.
- Action Space: Absolute angular positions of the Adroit hand joints.
- State Observation Space: The angular position of the finger joints, the pose of the palm of the hand, as well as the pose of the real pen and target goal.
- Visual Observation Space: one 128x128 RGB image from a third-person view camera.

**Adroit Hammer**

- Overall Description: The environment is based on the Adroit manipulation platform, a 28 degree of freedom system which consists of a 24 degrees of freedom ShadowHand and a 4 degree of freedom arm. The task to be completed consists on picking up a hammer with and drive a nail into a board.
- Task Difficulty: The nail position is randomized and has dry friction capable of absorbing up to 15N force.
- Object Variations: No object variations.
- Action Space: Absolute angular positions of the Adroit hand joints.
- State Observation Space: The angular position of the finger joints, the pose of the palm of the hand, the pose of the hammer and nail, and external forces on the nail.
- Visual Observation Space: one 128x128 RGB image from a third-person view camera.

**Adroit Relocate**

- Overall Description: The environment is based on the Adroit manipulation platform, a 30 degree of freedom system which consists of a 24 degrees of freedom ShadowHand and a 6 degree of freedom arm. The task to be completed consists on moving the blue ball to the green target.
- Task Difficulty: The positions of the ball and target are randomized over the entire workspace.
- Object Variations: No object variations.
- Action Space: Absolute angular positions of the Adroit hand joints.
- State Observation Space: The angular position of the finger joints, the pose of the palm of the hand, as well as kinematic information about the ball and target.
- Visual Observation Space: one 128x128 RGB image from a third-person view camera.

Table 2: We consider 8 continuous control tasks from 2 benchmarks. We list important task details below.

| Task | State Observation Dim | Action Dim | Max Episode Step |
|---|---|---|---|
| ManiSkill: StackCube | 55 | 4 | 200 |
| ManiSkill: PegInsertionSide | 50 | 7 | 200 |
| ManiSkill: TurnFaucet | 43 | 7 | 200 |
| ManiSkill: PushChair | 131 | 20 | 200 |
| Adroit: Door | 39 | 28 | 300 |
| Adroit: Pen | 46 | 24 | 200 |
| Adroit: Hammer | 46 | 26 | 400 |
| Adroit: Relocate | 39 | 30 | 400 |

## A.2 DEMONSTRATIONS

This subsection provides the details of demonstrations used in our experiments. See Table 3. ManiSkill demonstrations are provided in Gu et al. (2023), and Adroit demonstrations are provided in Rajeswaran et al. (2017).

Table 3: We list the number of demonstrations and corresponding generation methods below.

| Task | Num of Demo Trajectories | Generation Method |
|---|---|---|
| ManiSkill: StackCube | 1000 | Task and Motion Planning |
| ManiSkill: PegInsertionSide | 1000 | Task and Motion Planning |
| ManiSkill: TurnFaucet | 1000 | Model Predictive Control |
| ManiSkill: PushChair | 1000 | Reinforcement Learning |
| Adroit: Door | 25 | Human Teleoperation |
| Adroit: Pen | 25 | Human Teleoperation |
| Adroit: Hammer | 25 | Human Teleoperation |
| Adroit: Relocate | 25 | Human Teleoperation |

# B  IMPLEMENTATION DETAILS

## B.1  BASE POLICIES

We experiment with two state-of-the-art imitation learning models: Behavior Transformer (Shafiullah et al., 2022) and Diffusion Policy (Chi et al., 2023).

### B.1.1  BEHAVIOR TRANSFORMER

We follow the setup of Behavior Transformer in the original paper (Shafiullah et al., 2022). The architecture hyperparameters are included in Table 4, and the training hyperparameters are included in Table 5.

Table 4: We list the important architecture hyperparameters of Behavior Transformer used in our experiments.

| Hyperparameter | Value |
|---|---|
| Context Window | 10/20 |
| Num Clusters | 4/8 |
| Num Layers | 4 |
| Num Heads | 4 |
| Embedding Dimensions | 128 |
| Trainable Parameters | $\sim$ 1 Million |

Table 5: We list the important training hyperparameters of Behavior Transformer in ManiSkill and Adroit tasks below.

| Hyperparameter | Value (ManiSkill) | Value (Adroit) |
|---|---|---|
| Num of Gradient Steps | 200000 | 5000 |
| Batch Size | 2048 | 2048 |
| Learning Rate | 1e-4 | 1e-4 |
| Optimizer | AdamW Optimizer | AdamW Optimizer |

### B.1.2 DIFFUSION POLICY

We follow the setup of U-Net version of Diffusion Policy in the original paper (Chi et al., 2023). The architecture hyperparameters are includes in Table 6, and the training hyperparameters are included in Table 7.

Table 6: We list the important architecture hyperparameters of Diffusion Policy used in our experiments.

| Hyperparameter | Value |
|---|---|
| Observation Horizon | 2 |
| Action Horizon | 4 |
| Prediction Horizon | 16 |
| Embedding Dimensions | 64 |
| Downsampling Dimensions | 256, 512, 1024 |
| Trainable Parameters | $\sim$ 4 Million |

Table 7: We list the important training hyperparameters of Diffusion Policy in ManiSkill and Adroit tasks below.

| Hyperparameter | Value (ManiSkill) | Value (Adroit) |
|---|---|---|
| Gradient Steps | 200000 | 200000 |
| Batch Size | 1024 | 1024 |
| Learning Rate | 1e-4 | 1e-4 |
| Optimizer | AdamW Optimizer | AdamW Optimizer |

### B.1.3 CHECKPOINT SELECTION

We evaluate the base policy for 50 episodes every 50k environment steps during training. We select the checkpoint with the highest evaluation success rate and improve it afterward.

### B.2 POLICY DECORATOR (OUR APPROACH)

Policy Decorator framework introduces two key hyperparameters: $H$ **in Progressive Exploration Schedule** and **Bound $\alpha$ of Residual Actions**. We list the values of these two key hyperparameters across all tasks in the table below. Both of them are not too difficult to tune. *We typically set $\alpha$ close to the action scale observed in the demonstration dataset and make minor adjustments. $H$ has a wide workable range, and using a large $H$ is generally a safe choice if sample efficiency is not the primary concern.* See Section 5.4.2 for more discussion on the influence of these two hyperparameters.

Table 8: The values of $H$ in Progressive Exploration Schedule and Bound $\alpha$ of Residual Actions across all tasks.

| Task | $H$ | $\alpha$ |
|---|---|---|
| ManiSkill: StackCube (BeT, state) | 1M | 0.03 |
| ManiSkill: PegInsertionSide (BeT, state) | 8M | 0.3 |
| ManiSkill: TurnFaucet (BeT, state) | 500K | 0.2 |
| ManiSkill: PushChair (BeT, state) | 4M | 0.2 |
| Adroit: Door (BeT, state) | 100K | 0.3 |
| Adroit: Pen (BeT, state) | 100K | 0.3 |
| Adroit: Hammer (BeT, state) | 100K | 0.3 |
| Adroit: Relocate (BeT, state) | 100K | 0.2 |
| | | |
| ManiSkill: PegInsertionSide (Diffusion Policy, state) | 30K | 0.1 |
| ManiSkill: TurnFaucet (Diffusion Policy, state) | 100K | 0.1 |
| ManiSkill: PushChair (Diffusion Policy, state) | 300K | 0.2 |
| Adroit: Pen (Diffusion Policy, state) | 100K | 0.2 |
| Adroit: Hammer (Diffusion Policy, state) | 100K | 0.1 |
| Adroit: Relocate (Diffusion Policy, state) | 300K | 0.1 |
| | | |
| ManiSkill: TurnFaucet (Diffusion Policy, visual) | 30K | 0.05 |
| ManiSkill: PushChair (Diffusion Policy, visual) | 100K | 0.2 |
| Adroit: Door (Diffusion Policy, visual) | 1M | 0.1 |
| Adroit Pen (Diffusion Policy, visual) | 100K | 0.8 |

### B.3 IMPORTANT SHARED HYPERPRAMETERS AMONG POLICY DECORATOR AND OTHER BASELINES

As all baselines use SAC as the backbone RL algorithm, we include some important shared hyperparameters used among the Policy Decorator and baselines in our experiments. See the Table 9 for more details.

Table 9: We list the important shared hyperparameters among Policy Decorator and other baselines in ManiSkill and Adroit tasks below.

| Hyperparameter | Value (ManiSkill) | Value (Adroit) |
|---|---|---|
| Gamma | 0.97/0.9 | 0.97 |
| Batch Size | 1024 | 1024 |
| Learning Rate | 1e-4 | 1e-4 |
| Policy Update Frequency | 1 | 1 |
| Training Frequency | 64 | 64 |
| Update-to-data Ratio | 0.25 | 0.25 |
| Target Network Update Frequency | 1 | 1 |
| Tau | 0.01 | 0.01 |
| Learning Starts | 8000 | 8000 |

### B.4 ENABLE RL FINE-TUNING ON BASE POLICIES

As discussed in the introduction, many state-of-the-art imitation learning models are non-trivial to fine-tune using RL. In this section, we describe the modifications we made to the base policies to enable our fine-tuning-based baselines. Note that our approach does not require such modifications, which were implemented solely for the baselines.

### B.4.1 SAC FOR BEHAVIOR TRANSFORMER

Behavior Transformer (BeT) represents an action as a combination of a cluster center (predicted by a classification head) and an offset (predicted by a regression head). The action cluster centers are determined by k means, which is non-differentiable, thus we only fine-tune the offset using RL gradients.

**Special Modifications on BeT**    The SAC actor network uses a `tanh` layer to constrain its output space to $(-1, 1)$. However, the offset regression head of the Behavior Transformer does not have such a bounded output space. Therefore, we maintain the clustering process in BeT within the regular action space while adding an `tanh` layer to the regression head. The Behavior Transformers used in the fine-tuning-based baselines are trained with this `tanh` layer.

However, due to the bounded range of the `tanh` function (-1, 1), certain action values cannot be represented within this range, such as gripper actions that are exactly 1. To avoid numerical issues, we implement action rescaling. Specifically, in ManiSkill, the gripper dimension (last action dimension) is multiplied by 0.3, while in Adroit, all actions are multiplied by 0.5. These actions are then rescaled back to their original range after passing through the `tanh` function.

The modified BeT achieves similar performance in evaluations, thus enabling fair comparison. See Table 10 for the evaluation success rates of BeT and BeT modified version in ManiSkill and Adroit tasks.

Following the general paradigm of fine-tuning GPT-based models in natural language processing, we add LoRA to all attention layers and final regression heads.

Table 10: We list the evaluation success rate of BeT and BeT modified version in ManiSkill and Adroit tasks. BeT modified version is used in fine-tuning baselines, and original BeT is used in Policy Decorator and non-fine-tuning baselines.

| Task | BeT | BeT modified version |
|------|-----|----------------------|
| ManiSkill: StackCube (state) | 71% | 67% |
| ManiSkill: PegInsertionSide (state) | 15% | 13% |
| ManiSkill: TurnFaucet (state) | 41% | 35% |
| ManiSkill: PushChair (state) | 18% | 23% |
| Adroit: Door (state) | 78% | 77% |
| Adroit: Pen (state) | 65% | 63% |
| Adroit: Hammer (state) | 23% | 21% |
| Adroit: Relocate (state) | 20% | 13% |

**Setup on SAC**    We use SAC as our primary fine-tuning algorithm for the Behavior Transformer, with the actor initialized using a pre-trained Behavior Transformer and an MLP as the Q function. See Appendix F.5.1 for a discussion on the architecture choice of Q function.

### B.4.2 DIPO FOR DIFFUSION POLICY

**Special Modifications on DIPO**    DIPO employs action gradients to optimize actions and converts online training into supervised learning (more details in H.2). While the original DIPO uses a naive diffusion model as the actor, we need to integrate DIPO with the diffusion policy. Since the Diffusion Policy implements receding horizon control, it predicts more actions than are actually executed. During RL training, feedback is only available for executed actions. Therefore, we apply action gradients to update the executed actions while maintaining the predicted but unexecuted actions unchanged.

Following the general paradigm of fine-tining diffusion-based models in visual, we add LoRA to all layers of diffusion policy.

## B.5 BASELINES

In our experiments, we compare Policy Decorator with several strong baseline methods. The following section provides implementation details for these baseline approaches.

### B.5.1 FINE-TUNING METHODS

**Basic RL** See Appendix B.4.

**ROT** (Haldar et al., 2023a) is an online fine-tuning algorithm that fine-tunes a pre-trained base policy using behavior cloning (BC) regularization with adaptive Q-filtering and optimal transport (OT) rewards. We use a pre-trained Behavior Transformer as the base policy. For Behavior Cloning regularization, we allow BeT to output the entire window of actions and apply the regularization accordingly. In experiments involving state observations, the optimal transport (OT) rewards are computed using a 'trunk' network within the value function, which consists of a single-layer neural network. In contrast, for experiments with visual observations, the OT rewards are computed directly using the visual encoder network. The other experimental setup follows B.4.1.

**RLPD** (Ball et al., 2023) is a state-of-the-art online learn-from-demo method that enhances the vanilla SACfd with critic layer normalization, symmetric sampling, and sample-efficient RL (Q ensemble + high UTD). In our implementation, we maintain one offline buffer, which includes demonstration data, and one online buffer, which contains online data. For online updates, we sample 50% batch from the offline buffer and 50% batch from the online buffer. We omit the sample-efficient RL (Q ensemble + high UTD) due to the significant training costs associated with these components and to ensure a fair comparison with other methods. The omitted component pursues extreme sample efficiency at the cost of significantly increased wall-clock training time, which is impractical, especially when fine-tuning a large model. The other experiment setup follows B.4.1.

**Cal-QL** (Nakamoto et al., 2024) is an offline RL online fine-tuning method that "calibrates" the Q function of vanilla CQL. We pre-train a Q function using Cal-QL in the offline stage and then use SAC for fine-tuning in the online stage with this pre-trained value function. We opted for this offline-to-online strategy because, in the online stage of the original Cal-QL paper, calculating the critic loss requires querying the actor 20 times. This process is time-intensive, especially considering that the actor is initialized as a large base model. The performance of curve C in Fig. 22 demonstrates the effectiveness of this strategy. See F.3 for more discussion. In the offline stage, we use the pre-trained BeT with gradients open as the actor and an MLP as the critic. In the online stage, we use the pre-trained BeT as the actor and the offline-trained MLP as the critic. The other experiment setup follows B.4.1.

### B.5.2 NON-FINE-TUNING METHODS

**JSRL** (Uchendu et al., 2023) is a curriculum learning algorithm that uses an expert teacher policy to guide the student policy. In our setting, we use a pre-trained large policy (BeT or diffusion policy) as the guiding policy and an MLP as the online actor. The initial jump start steps are the average length of success trajectories in 100 evaluations of the pre-trained base policy. Following the setup in the original paper, we maintain a moving window of evaluation success rate and best moving average success rate. If the current moving evaluation success rate is within the range of [best moving average - tolerance, best moving average + tolerance], then we go 10 steps backward.

**Residual RL** (Johannink et al., 2019) learns a residual policy in an entirely uncontrolled manner. In our experiments, We use a pre-trained large policy as the base policy and a small MLP as the online residual actor. We follow the setting in the original paper that in online interactions, final action = base action + online residual action.

**FISH** (Haldar et al., 2023b) builds upon Residual RL by incorporating a non-parametric nearest neighbor search VINN policy (Pari et al., 2021) and learning an online offset actor with optimal transport rewards. In our experiments, we use a GPT backbone as the representation network for BeT experiments, a FiLM encoder (Perez et al., 2018) for diffusion state observation mode experiments, and a visual encoder for visual observation mode experiments. See Appendix G.2 for the performance of VINN policy.

## C   ADDITIONAL BASELINES

### C.1   COMPARISON WITH DPPO

#### C.1.1   SETUP

DPPO (Ren et al., 2024), a concurrent work, successfully fine-tunes diffusion policies using PPO, achieving state-of-the-art performance. Key tricks include fine-tuning only the last few denoising steps and fine-tuning DDIM sampling. We conducted preliminary experiments comparing our approach with DPPO **on their tasks**. Even if DPPO is carefully tuned on their tasks, we are still able to beat it.

Specifically, we applied Policy Decorator (our approach) to the **two most challenging robotic manipulation tasks in their paper**: Square and Transport. We used the Diffusion Policy checkpoints provided by the DPPO paper as our base policies.

#### C.1.2   RESULTS

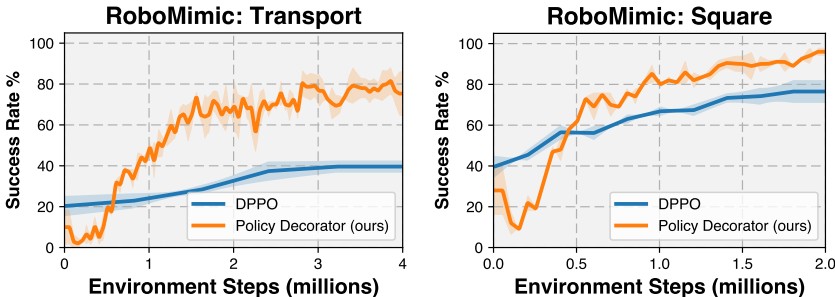

Figure 13: Policy Decorator (ours) vs. DPPO on two most challenging robotic manipulation tasks in the DPPO paper: Square and Transport.

As shown in Fig. 13, our approach outperforms DPPO on these two challenging tasks, especially on the Transport task. According to Figure 5 in the DPPO paper, DPPO requires approximately 16 million steps to converge to 80%+ success rate on the Transport task. In contrast, our Policy Decorator achieves this performance in only 4 million steps, demonstrating a **nearly 4x improvement in sample efficiency**.

**Implementation details:** We noticed that DPPO uses a fixed episode length without early termination upon success signals. Empirically, this setup may negatively impact the sample efficiency of RL algorithms, as transitions after task completion contribute minimally to learning. We therefore implement early task termination upon success signals in our experiments, which offers a more practical setup. Results using DDPO's original setup are presented in Fig. 14.

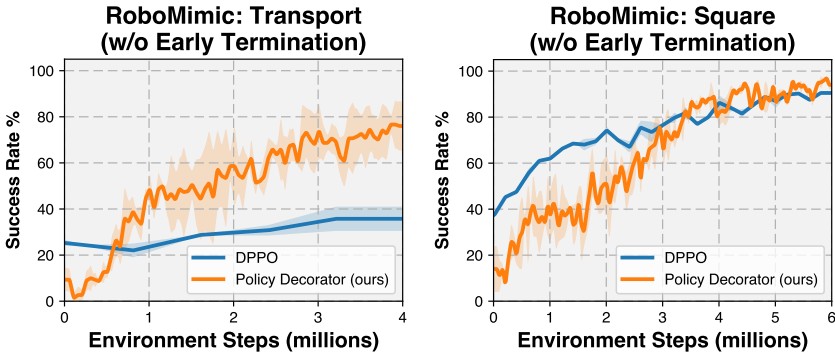

Figure 14: Similar to Fig. 13, but without early termination upon success signals.

### C.1.3 SUMMARY

**These experiments demonstrate that our method outperforms DPPO on challenging robotic manipulation tasks.** It is crucial to note that our approach is model-agnostic, whereas DPPO is restricted to a specific case of Diffusion Policy (where all predicted actions are executed in the environment, which is not the typical implementation of Diffusion Policy).

## D    ADDITIONAL ABLATION STUDIES

This section includes additional ablation studies results about base policies, low-performing checkpoints, and PPO. In detail, Section D.1 discusses Policy Decorator also works with other types of base policies (e.g., MLP, RNN, and CNN); Section D.2 demonstrates that Policy Decorator stays effective in improving low-performing BeT checkpoints; Section D.3 indicates that Policy Decorator is compatible with PPO as backbone RL algorithm.

### D.1    ADDITIONAL BASE POLICIES

To demonstrate Policy Decorator's versatility across different base policy architectures, we conduct experiments using simpler models such as MLP, BC-RNN (Mandlekar et al., 2021), and CNN. Fig. 15 shows that Policy Decorator significantly improves the performance of MLP, BC-RNN, and CNN policies through online interactions.

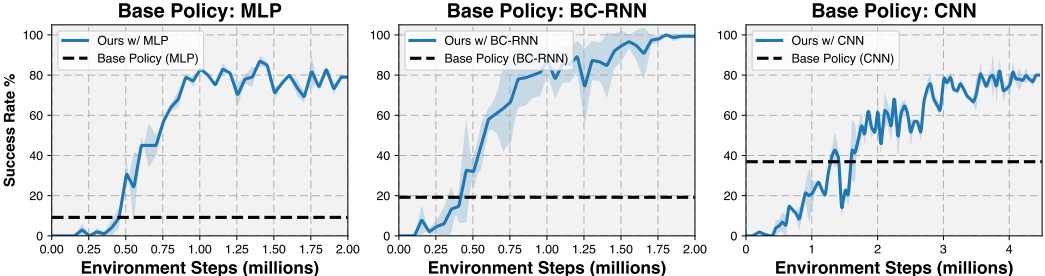

Figure 15: Policy Decorator with more base policies (MLP, BC-RNN, CNN) on TurnFaucet task.

### D.2    USING OTHER CHECKPOINTS OF BASE POLICIES

As we claim that Policy Decorator is model-agnostic and is versatile to all types of base policies, it is necessary to demonstrate that it not only improves well-trained base policy but also improves low-performing checkpoints of base policy. Fig. 16 shows that the Policy Decorator achieves a substantial improvement in the low-performance BeT checkpoint.

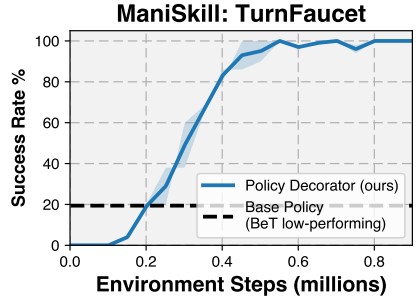

Figure 16: Policy Decorator with a low-performance BeT checkpoint.

## D.3 CHANGE BACKBONE RL ALGORITHM TO PPO

While our experiments utilize SAC as the backbone RL algorithm due to its sample efficiency, Policy Decorator is also compatible with various RL algorithms. In this experiment, we replace SAC with PPO (Schulman et al., 2017) across our method, the RL fine-tuning baseline, and the residual RL baseline. As demonstrated in Fig. 17, Policy Decorator with PPO successfully improves the base policy's performance and significantly outperforms all baseline methods.

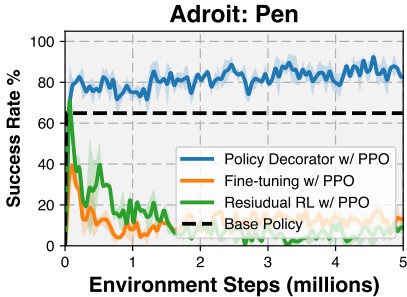

Figure 17: Use PPO as the backbone RL algorithm in our method, RL fine-tuning, and Residual RL.

## D.4 ADDITIONAL BASELINE (GAIL + MLP)

As mentioned in Chi et al. (2023), simple architectures like MLP cannot capture the multi-modality of data. To further demonstrate this point, we have now implemented and tested a GAIL + MLP baseline. The results in Fig. 18 show that this baseline achieves 0% success rate on StackCube and about 20% success rate on TurnFaucet after 3M environment interactions. These results are expected given that the demonstrations were collected task and motion planning (for StackCube) and model predictive control (for TurnFaucet) - resulting in naturally multi-modal distributions. These results suggest that simple MLPs may be insufficient for capturing multi-modal distributions and highlight the need for large policy models for effectively utilizing multi-modal demonstrations.

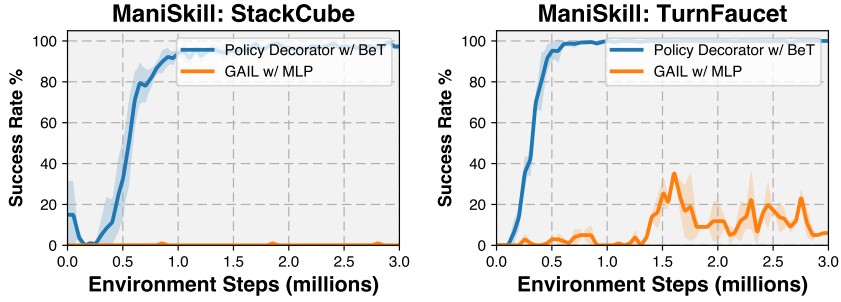

Figure 18: Comparison of GAIL + MLP and Policy Decorator.

Additionally, in offline imitation learning scenarios, MLP also performs significantly worse than large policy models, as shown in the table below:

These results together demonstrate that simple MLPs are insufficient for capturing multi-modal distributions and highlight the need for large policy models to effectively utilize multi-modal demonstrations.

Table 11: Performance comparison across tasks (StackCube and TurnFaucet).

| Model | StackCube | TurnFaucet |
|---|---|---|
| MLP | 0% | 10% |
| Behavior Transformer | 71% | 41% |
| Diffusion Policy | 99% | 55% |

# E    IMPORTANT DESIGN CHOICES

This section presents ablation results on a few key design choices, including the inputs for the residual policy and the inputs for the critic.

## E.1    INPUT OF RESIDUAL POLICY

The residual policy can receive input in the form of either observation alone or both observation and action from the base policy. Our experiments indicate that using only observation typically produces better results, as illustrated in Fig. 19.

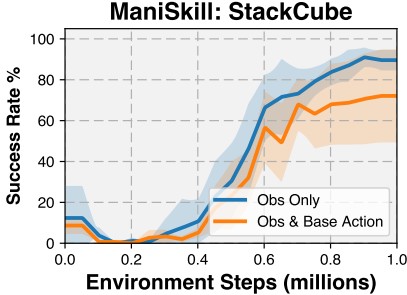

Figure 19: Different variants of input of residual policy.

## E.2    INPUT OF CRITIC

In SAC, the critic $Q(s, a)$ takes an action as input, and there are several design choices regarding this action: we can use 1) the sum of the base action and residual action; 2) the concatenation of both; or 3) the residual action alone. Based on our experiments shown in Fig. 20, using the sum of both actions yields the best performance.

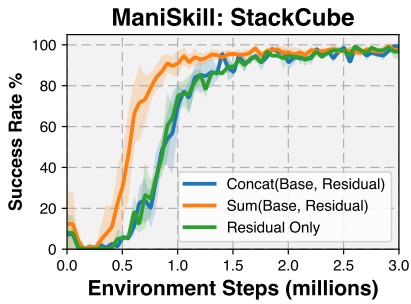

Figure 20: Different variants of input of critic.

## F  FAILURE OF FINE-TUNING BASELINES

In this section, we analyze the poor performance of fine-tuning baselines in our experiments. **We provide an overall explanation for these failures in Sec. F.1. Then, Sec. F.2, F.3, and F.4 offer illustrative experiments supporting the arguments presented in Sec. F.1.** Finally, Sec. F.5 presents some additional ablation studies on design choices in fine-tuning baselines, demonstrating our careful tuning of baseline implementations to achieve better performance.

### F.1  OVERALL EXPLANATION

Even if we have selected the strongest learning-from-demo methods, most of them are still not specifically designed for fine-tuning, and they do not intentionally prevent the unlearning of the base model, i.e., the performance can drop significantly at the very beginning of training. This phenomenon has also been discussed in Nakamoto et al. (2024). According to our observations, we believe that performance degradation is probably due to the following two reasons:

1. **Random Critic Initialization:** We believe the randomly initialized critic network cannot provide meaningful gradients to guide the policy. Such a noisy gradient can easily cause the policy to deviate significantly from the initial weights. Once the unlearning happens, it becomes very hard to relearn the policy since it cannot get the sparse reward signal anymore. Sec. F.2 presents an illustrative experiment to show this policy degradation with randomly initialized critic. On the other hand, Cal-QL (Nakamoto et al., 2024) can theoretically learn a critic from offline data. However, our empirical results indicate that when trained purely on demonstration data without negative trajectories, the learned critic does not significantly improve online fine-tuning. This performance degradation during Cal-QL online training aligns with observations reported by (Yang et al., 2023a). Experimental evidence supporting this analysis is presented in Sec. F.3.

2. **Long Task Horizon:** Long task horizon also significantly increases the difficulty of fine-tuning, particularly in sparse reward settings. As the task horizon increases, the agent's likelihood of discovering sparse rewards through random exploration diminishes. Additionally, the sparse reward signal requires more time to propagate through longer trajectories. The experiments presented in Sec. F.4 empirically validate that the long task horizon is a key factor contributing to the failure of fine-tuning baselines.

### F.2  POLICY DEGRADATION WITH RANDOM INITIALIZED CRITIC

This section presents illustrative experiments demonstrating how updating the base policy with a randomly initialized critic function $Q(s, a)$ results in significant deviations from its original trajectory.

In the StackCube task, a robot arm must pick up a red cube and stack it on a green cube. Initially, a pre-trained base policy (Behavior Transformer) successfully grasps the red cube and accurately places it on the green cube, as shown in this video.

After fine-tuning the base policy with a randomly initialized critic for 100 gradient steps, the policy begins to deviate slightly from the original trajectory, as shown in this video. While still able to grasp the red cube, it fails to precisely place it on the green cube.

Following an additional 100 updates (200 total), the base policy deviates further from the original trajectory, struggling to effectively grasp the red cube, as shown in this video.

**In summary, these experiments suggest that fine-tuning the base policy with a randomly initialized critic can lead to unlearning. Once unlearning occurs, it becomes very hard to relearn the policy since it cannot get the sparse reward signal anymore.**

### F.3  PRE-TRAINING CRITIC ON DEMO-ONLY DATASET DOES NOT HELP

Cal-QL (Nakamoto et al., 2024), a state-of-the-art offline RL method, aims to pre-train a critic for efficient online fine-tuning. Our experiments show that pre-training a critic using Cal-QL on demonstration-only datasets (without negative experiences) provides limited benefits for online

fine-tuning, as illustrated in Fig. 21. **This section presents experiments explaining why it does not help and validates the correctness of our Cal-QL baseline results.**

The original Cal-QL paper reported much better results on Adroit tasks compared to our Cal-QL baseline. We believe this discrepancy is mainly due to differences in experimental setups:

1. **Offline Dataset:** The original Cal-QL paper uses an offline dataset consisting of 25 human teleoperation demonstrations and additional trajectories from a BC policy. Our Cal-QL baseline uses only 25 human demonstrations, ensuring fair comparison with other learning-from-demo baselines that only utilize demonstrations. We also made this assumption in Sec. 3.

2. **Actor Architecture:** The original Cal-QL paper employs a small MLP as the actor, while we use a pre-trained Behavior Transformer (BeT) to align with our goal of improving the pre-trained base policy.

3. **Online Algorithm:** The original Cal-QL paper uses Cal-QL algorithm in both offline and online stage. However, computing critic loss in Cal-QL algorithm requires querying the actor 20 times in each update, which is extremely time-consuming given that the actor is a large model in our settings. Therefore, we use SAC in the online phase instead of Cal-QL.

To verify whether these setup differences cause the divergent results, we designed the following experimental setups for Cal-QL, **interpolating between the original setup and ours:**

- A: Small MLP actor + Mixed dataset + online Cal-QL (Cal-QL's original setting)
- B: Small MLP actor + Demo-only dataset + online Cal-QL
- C: Small MLP actor + Demo-only dataset + online SAC
- D: Large GPT actor + Demo-only dataset + online SAC
- E: BeT actor + Demo-only dataset + online SAC (the setup used in our experiments)

The experimental results of these setups are shown in Fig. 22. **In Cal-QL's paper, they only report the results up to 300k steps, and our curve A perfectly matches the official results, which suggests that our implementation is correct.** Interestingly, Cal-QL exhibits instability when run for longer periods (e.g., 3M steps), even in its original setup. Comparing curve A and curve B illustrates Cal-QL's strong dependence on a large, diverse dataset comprising both demonstrations and negative trajectories. Cal-QL's sample efficiency deteriorates a lot when the offline dataset is limited to a few demonstrations without negative trajectories. The comparison between curve B and curve C demonstrates that while using SAC as an online algorithm results in slightly reduced sample efficiency, it still achieves 90%+ success rates. This trade-off suggests that *sacrificing a little bit of sample efficiency is acceptable in exchange for significant wall-clock time savings*. The comparison between curve C and curve D illustrates that a large GPT actor can also negatively impact Cal-QL's performance. Curve D and curve E demonstrate that using a pre-trained BeT outperforms a randomly initialized GPT, which is expected.

**In conclusion, the divergent results between Cal-QL's original paper and our baseline can be attributed to different experimental setups. Our results are validated and reliable.**

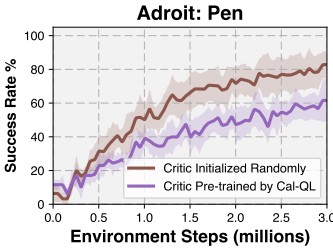

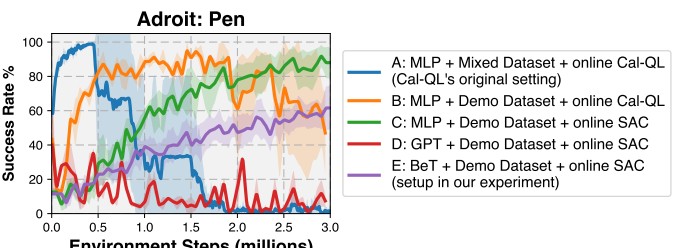

Figure 21: Pre-training a critic by Cal-QL on demo-only datasets does not help online fine-tuning.

Figure 22: To verify whether the setup differences cause the divergent results, we designed different experimental setups for Cal-QL, interpolating between the original setup and ours.

## F.4    Long Task Horizon Makes Fine-tuning Hard

This section presents experiments exploring how task horizon affects the fine-tuning of the base policy.

In the TurnFaucet task, no fine-tuning baselines achieve non-zero success rates. To shorten the effective task horizon, we roll out the pre-trained base policy (Behavior Transformer) for a specific number of steps (40, 100, or 120) in each episode. This approach likely brings the agent closer to success, thus shortening the effective task horizon. We then perform regular RL fine-tuning for the remaining steps of an episode.

Fig. 23 demonstrates that shortening the task horizon by 100 steps results in a significant improvement, while reducing it by 120 steps achieves a 100% success rate. This experiment clearly shows that the **long task horizon is a major factor in fine-tuning failure, and reducing the task horizon substantially eases RL fine-tuning difficulties.**

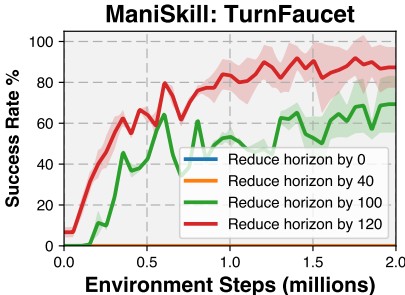

Figure 23: Fine-tuning Behavior Transformer using SAC with different effective task horizons.

## F.5    Ablation Study on Design Choices in Fine-tuning Baselines

This section contains ablation studies on some design choices in fine-tuning-based baselines. In detail, Section F.5.1 discusses different choices of Q function architecture, while Section F.5.2 illustrates the effects of using warmstart in Q function training.

### F.5.1    Architecture of Q Function

The architecture of the Q function can be important in designing fine-tuning baselines. We have three options for the Q function:

1. Use a randomly initialized MLP

2. Use a randomly initialized GPT

3. Use the pre-trained GPT backbone from the actor, and add a randomly initialized value head

As shown in Fig. 24, we experimented with all the aforementioned Q-function architectures in SAC fine-tuning experiments. The results indicate that fine-tuning with an MLP Q-function slightly improves the base policy, whereas fine-tuning with the other two Q-function architectures does not yield such improvements. Based on these observations, we chose to use the MLP Q-function in our fine-tuning baselines.

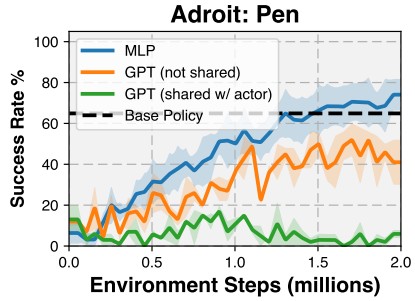

Figure 24: Fine-tuning with different critic architectures.

### F.5.2 EFFECT OF WARM-START IN Q FUNCTION TRAINING

Warm-starting Q function training is a common technique used to ensure the actor receives updates from a more reliable Q function. When designing our fine-tuning baselines, we explored this technique by training the critic for several steps before activating actor training. However, as shown in Fig. 25, this method leads to a substantial increase in alpha, the learnable entropy coefficient in SAC, resulting in an unstable explosion of the Q loss.

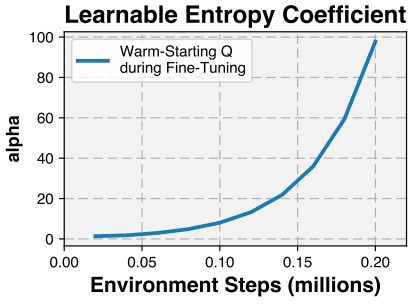
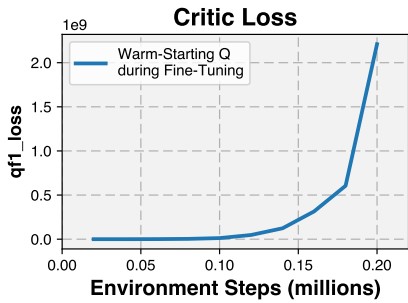

Figure 25: Critic warm start results in alpha (the learnable entropy coefficient in SAC) and critic loss explosion when auto entropy tuning is enabled.

Since the entropy coefficient alpha is exploding, it is natural to try freezing the entropy coefficient during warm-start and unfreezing it in subsequent fine-tuning. Results are shown in Fig. 26. From six independent runs, three of them still blow up upon unfreezing while the other three remained stable. This result indicates that this unfreezing strategy does not effectively address the training stability issue associated with warm-starting.

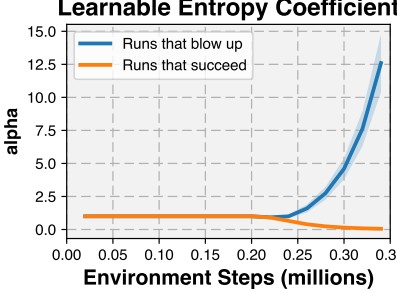
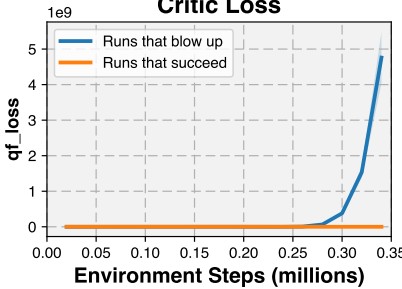

Figure 26: **Entropy coefficient and critic loss.** We fixed the entropy coefficient alpha during the warm-start phase (0.2M steps) and unfreeze it during fine-tuning. We merge six independent runs into two groups: three of them blow up while the other three remain stable.

We also tried fixing the alpha during the entire training process. As shown in Fig. 27, experimental results indicate that vanilla fine-tuning performs better than fine-tuning with warm-start and fixed alpha. Further analysis showed that using a fixed alpha leads to unstable critic training. **Consequently, we opted not to implement warm-start Q function training in our fine-tuning baselines.**

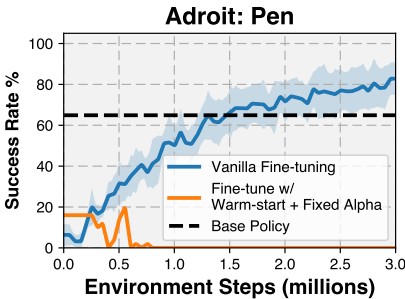

Figure 27: Fine-tuning with warm-starting and fixed alpha.

## G    FAILURE OF NON-FINE-TUNING BASELINES

In this section, we analyze the poor performance of non-fine-tuning baselines in our experiments. We discuss the failure of vanilla Residual RL in Section G.1. We provide the explanations of the failure of FISH in Section G.2.

### G.1    FAILURE OF VANILLA RESIDUAL RL

The residual RL baseline uses identical settings to our method, except for the controlled exploration module. The failure of residual RL mainly stems from two key issues:

1. During early training stages, random residual actions cause significant deviations from the base policy's trajectory. These deviations prevent the agent from receiving success signals necessary for guiding learning (see this video for an example).
2. The residual policy is designed to make minor corrections to the base policy. However, without explicit constraints, the magnitude of residual actions often exceeds that of the base policy actions, destroying the base policy's behavior.

Our ablation study (Fig. 10 and 11) supports these observations. As we progressively remove controlled exploration strategies (reducing $H$ to 0 or increasing $\alpha$ to 1), our method approaches vanilla residual RL, leading to degraded performance.

### G.2    FAILURE OF FISH

The failure of FISH primarily stems from the significantly poor performance of its non-parametric base VINN policy. The performance metrics of the VINN base policies are shown below.

Table 12: The performance of VINN base policy using GPT backbone from BeT (state observation).

| Task | Success Rate |
| --- | --- |
| ManiSkill: StackCube | 0% |
| ManiSkill: PegInsertionSide | 0% |
| ManiSkill: TurnFaucet | 1% |
| ManiSkill: PushChair | 0% |
| Adroit: Door | 12% |
| Adroit: Pen | 16% |
| Adroit: Hammer | 0% |
| Adroit: Relocate | 2% |

Table 13: The performance of VINN base policy using FiLM encoder from Diffusion Policy under state observation.

| Task | Success Rate |
|---|---|
| ManiSkill: PegInsertionSide | 0% |
| ManiSkill: TurnFaucet | 0% |
| ManiSkill: PushChair | 0% |
| Adroit: Pen | 16% |
| Adroit: Hammer | 0% |
| Adroit: Relocate | 0% |

Table 14: The performance of VINN base policy using visual encoder from Diffusion Policy under visual observation.

| Task | Success Rate |
|---|---|
| ManiSkill: TurnFaucet | 0% |
| ManiSkill: PushChair | 0% |
| Adroit: Door | 0% |
| Adroit: Pen | 8% |

## H   FINE-TUNING DIFFUSION POLICY USING RL

### H.1   WHY FINE-TUNING DIFFUSION POLICY USING RL IS NON-TRIVIAL

Diffusion Models (Ho et al., 2020) and their applications in robotic control (Chi et al., 2023; Janner et al., 2022; Ajay et al., 2022) have traditionally been trained using supervised learning, where ground truth labels (e.g., images, actions) are required to supervise the denoising process.

Recently, novel approaches (Fan & Lee, 2023; Black et al., 2023; Uehara et al., 2024) have emerged, proposing the use of reinforcement learning (RL) to train diffusion models. The high-level idea involves modeling the denoising process as a Markov Decision Process (MDP) and assigning rewards based on the quality of the final denoised samples. This allows RL gradients to be backpropagated through the **inference process**, updating the model weights accordingly. This training paradigm represents a significant departure from conventional diffusion model training methods and **may face challenges when the number of denoising steps is large**. To date, these methods have primarily been applied in the domains of **image generation, molecule design, and DNA synthesis**.

However, **this training paradigm does not directly transfer to robotic control problems, particularly in sparse reward tasks**. As discussed in Ren et al. (2024), fine-tuning diffusion models in robotic control can be viewed as a "two-layer" MDP, where a complete denoising process with hundreds of steps represents a single decision step in the robotic control MDP. For example, if a robotic task requires 200 decision steps (actions) to complete, and a diffusion model uses 100 denoising steps to generate a decision (action), the reward in a sparse-reward robotic control task would be received only *every 20,000 denoising steps*. This presents a significantly greater challenge than training a diffusion model to generate images using RL, where rewards are typically received *every 100 denoising steps* under the same assumptions.

### H.2   HOW "BASIC RL FOR DIFFUSION POLICY" BASELINE IS SELECTED

Despite the challenges in training diffusion policies for robotic control using RL, recent attempts have emerged. These can be broadly grouped into three categories. We will briefly explain each method and discuss the selection of the "Basic RL" baseline for fine-tuning diffusion policy.

**Converting RL into Supervised Learning** Methods in this category adhere to the conventional training recipe of the diffusion models, and try to define a "ground truth action label" for supervision. DIPO (Yang et al., 2023b) introduces "action gradient," using gradient descent on $Q(s, a)$ to estimate the optimal action for state $s$. **DIPO is selected as the basic RL algorithm in our experiments.** IDQL (Hansen-Estruch et al., 2023) constructs an implicit policy by reweighting samples from a diffusion-based policy, and using the implicit policy to supervise the training of the diffusion-based policy. We did not select it as the fine-tuning baseline for two reasons: 1) the training can be extremely slow especially with large base policies, because IDQL involves sampling the diffusion model multiple times (32 to 128 in their code) to compute the implicit policy; 2) as reported in its paper, IDQL performs worse than Cal-QL and RLPD, which are included in our baselines.

**Matching the Score to the Q Function** QSM (Psenka et al., 2023) aims to match the score $\Psi$ of the diffusion-based policy to the gradient of the Q function $\nabla_a Q^\Psi(s, a)$ using supervised learning. According to Ren et al. (2024), QSM performs poorly in robotic manipulation tasks, thus it is not considered a competitive baseline.

**Backpropagating RL Gradients Through the Inference Process** Methods in this category adapt the training recipe discussed in H.1 to robotic control tasks, employing additional techniques to make it work. The actor's training objective is to maximize $Q(s, a)$. Diffusion QL (Wang et al., 2022) represents a basic version of these methods, primarily used in offline RL settings. However, its online performance is poor, as reported by Ren et al. (2024). Consistency AC (Ding & Jin, 2023) distills diffusion models into consistency models, significantly shortening the gradient propagation path. Nevertheless, its offline-to-online performance, as reported in its own paper, is even worse than Diffusion QL, thus we do not consider it a competitive baseline.

DPPO (Ren et al., 2024), a very recent work, successfully fine-tunes diffusion policies using PPO, achieving state-of-the-art performance. Key tricks include fine-tuning only the last few denoising steps and fine-tuning DDIM sampling. We conducted preliminary experiments comparing our approach with DPPO *on their tasks*. Results indicate that our method significantly outperforms DPPO on their tasks. See Appendix C.1 for more details.

## I  FORWARD AND BACKWARD TIME BENCHMARK

Compared to fine-tuning the base policy, our Policy Decorator eliminates the backward pass computation of the base policy while retaining the forward pass. To demonstrate that the backward pass is indeed the dominant computational factor, we conducted benchmarks on the Behavior Transformer's backward pass (gradient update) and forward pass (inference) running times. Results are shown in Fig. 28.

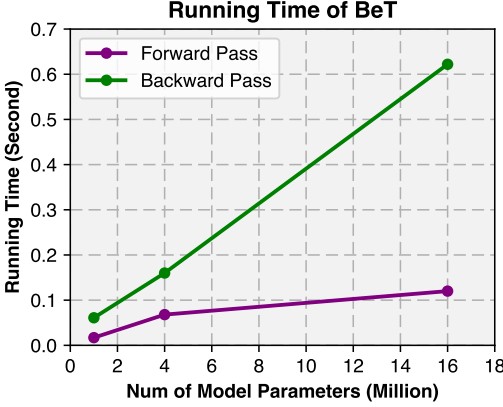

Figure 28: Running time comparison of forward and backward passes of the Behavior Transformer under different numbers of parameters.

The results demonstrate that BeT's forward pass is significantly faster than its backward pass, with this gap becoming more pronounced as model size increases. This confirms that the backward pass constitutes the major training time bottleneck.

**Implementation Details:**

- Batch Size: 1024
- GPU: NVIDIA GeForce RTX 2080 Ti
- Results averaged over 100 independent runs

Additionally, we present the actual training wall-clock time comparison below, demonstrating that our Policy Decorator is indeed more time-efficient compared to naive fine-tuning.

Table 15: Training time of Policy Decorator and SAC fine-tuning on StackCube. The base policy is Behavior Transformer. 5M environment steps.

|  | **StackCube, BeT, 5M Env Steps** |
| --- | --- |
| Policy Decorator (ours) | 7h 23m |
| SAC Fine-tuning | 33h 52m |

## J    MULTI-MODALITY PROPERTY OF THE COMBINED POLICY

In this paper, applying a small residual action to correct a multi-modal base policy typically maintains its multi-modal property. We illustrate this point through both an illustrative example and a real case study from our experiments.

### J.1    ILLUSTRATIVE EXAMPLE

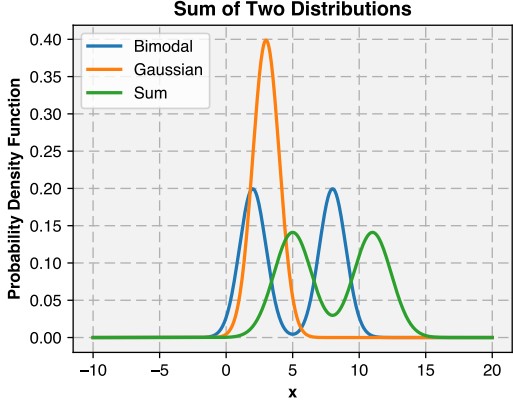

Figure 29: **Illustrative Example.** Adding a Gaussian distribution to a multi-modal distribution typically maintains its multi-modal property.

As demonstrated in Fig. 29, when a bimodal distribution (blue) is combined with a Gaussian distribution (orange), the sum distribution (green) still preserves its bimodal nature. This process effectively shifts the multi-modal distribution and adjusts the standard deviation of its modes. The multi-modal property is maintained as long as the Gaussian distribution's variance remains relatively small compared to the separation between modes.

**Implementation Notes:**

- The probability density function (PDF) of the bimodal distribution (blue):
$$f_{\text{Bimodal}}(x) = w_1 \cdot \mathcal{N}(x; \mu_1, \sigma_1^2) + w_2 \cdot \mathcal{N}(x; \mu_2, \sigma_2^2),$$

where $\mathcal{N}$ represents the Gaussian distribution.

- The PDF of the Gaussian distribution (orange):

$$f_{\text{Gaussian}}(x) = \mathcal{N}(x; \mu_3, \sigma_3^2).$$

- The PDF of the sum of the two distributions (green) can be computed analytically:

$$f_{\text{sum}}(x) = w_1 \cdot \mathcal{N}(x; \mu_4, \sigma_4^2) + w_2 \cdot \mathcal{N}(x; \mu_5, \sigma_5^2),$$

where:

$$\mu_4 = \mu_1 + \mu_3, \quad \sigma_4 = \sqrt{\sigma_1^2 + \sigma_3^2}, \quad \mu_5 = \mu_2 + \mu_3, \quad \sigma_5 = \sqrt{\sigma_2^2 + \sigma_3^2}.$$

- The parameters used in the plot are:

$$w_1 = 0.5, \quad w_2 = 0.5, \quad \mu_1 = 0.5, \quad \mu_2 = 0.5, \quad \mu_3 = 3, \quad \sigma_1 = 1, \quad \sigma_2 = 1, \quad \sigma_3 = 1.$$

### J.2 REAL CASE STUDY FROM OUR EXPERIMENTS

To demonstrate the preservation of multi-modality in practice, we visualize action distributions from a specific state in the ManiSkill StackCube task, using Behavior Transformer as the base policy. We sampled 1000 actions from both base and residual policies, then applied PCA dimensionality reduction for visualization purposes. We use histograms to visualize these action samples. The results are shown in Fig. 30.

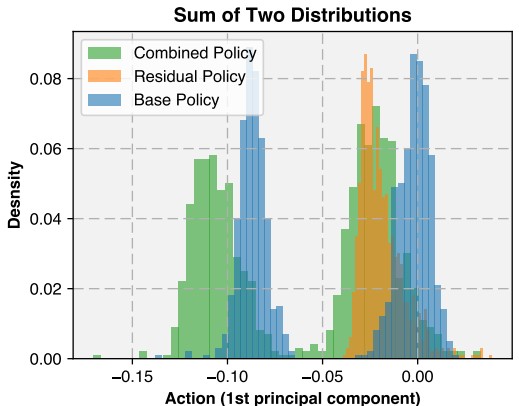

Figure 30: **Real Case Study from Our Experiments.** Applying a small residual action to correct a multi-modal base policy typically matains its multi-modal property.

We can see that the base policy exhibits a clear bimodal distribution. When combined with the residual policy, the sum distribution maintains its bimodal nature while exhibiting slight shifts in position and variance. Note that the residual policy here is actually a squashed Gaussian distribution (as per SAC (Haarnoja et al., 2018)) rather than a pure Gaussian, due to SAC's action bounds requirement. This practical example aligns well with our illustrative example, confirming that multi-modal property is preserved in our actual experiments.

