#### B.4.1 SAC FOR BEHAVIOR TRANSFORMER

**Special Modifications on BeT** Special adaptations relate to SAC's Gaussian Tanh Policy, which requires the actor backbone to output in the ATANH space of action rather than the regular space. This requirement complicates the initialization of the Behavior Transformer (BeT) as the actor backbone. Therefore, we allow the clustering process in BeT to operate in the regular action space, but the regression head outputs in the ATANH action space. The final action is then computed as:

$$\mathbf{a}_{\text{final}} = \text{arctanh}(\mathbf{a}_{\text{bin}}) + \mathbf{a}_{\text{regression output}}$$

Since the atanh function is defined between -1 and 1, some action dimensions (e.g., gripper actions) need to be scaled to avoid numerical issues. In ManiSkill, we multiply the gripper dimension (last action dimension) by 0.3; in Adroit, we multiply all actions by 0.5. The actions are rescaled back after going through tanh. Our BeT, specially modified for fine-tuning, achieves similar performance in evaluations in order to enable fair comparison. See Table 10 for evaluation success rate of BeT and BeT modified version in ManiSkill and Adroit tasks.

Following the general paradigm of fine-tuning GPT-based models in natural language processing, we add LoRA to all attention layers and final regression heads.

Table 10: We list the evaluation success rate of BeT and BeT modified version in ManiSkill and Adroit tasks. BeT modified version is used in fine-tuning baselines, and original BeT is used in Policy Decorator and non-fine-tuning baselines.

| Task | BeT | BeT modified version |
|---|---|---|
| ManiSkill: StackCube (state) | 71% | 67% |
| ManiSkill: PegInsertionSide (state) | 15% | 13% |
| ManiSkill: TurnFaucet (state) | 41% | 35% |
| ManiSkill: PushChair (state) | 18% | 23% |
| Adroit: Door (state) | 78% | 77% |
| Adroit: Pen (state) | 65% | 63% |
| Adroit: Hammer (state) | 23% | 21% |
| Adroit: Relocate (state) | 20% | 13% |

**Special Modifications on SAC** We use SAC as our primary fine-tuning algorithm for Behavior Transformer, with actor initialized using a pre-trained Behavior Transformer and a MLP as Q function. See Appendix F.5.1 for discussion on the architecture choice of Q function.

### B.4.2 DIPO FOR DIFFUSION POLICY

**Special Modifications on DIPO**    DIPO uses action gradients to optimize the actions, and convert online training to supervised learning, also refer to H.2. Since the Diffusion Policy employs a prediction horizon that exceeds the action horizon (receding horizon), during the DIPO training phase, we focus on optimizing only the first action horizon within the total prediction horizon using action gradients. This approach prevents dynamics inconsistencies that would arise from optimizing the remaining actions.

Following the general paradigm of fine-tining diffusion-based models in visual, we add LoRA to all layers of diffusion policy.

### B.5 BASELINES

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

## C    ADDITIONAL RESULTS OF POLICY DECORATOR

### C.1    THE PERFORMANCE OF RL FROM SCRATCH

The RL training from scratch baseline has been incorporated into Fig. 13. We only plot results on Adroit, as RL training from scratch achieves 0% success rate on ManiSkill tasks.

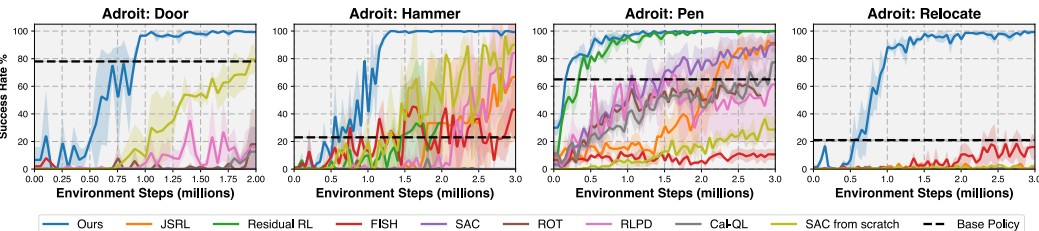

Figure 13: Add SAC (training from scratch) to Fig. 6. Results are only shown for Adroit tasks, as it achieves 0% success rate on all ManiSkill tasks with sparse reward.

### C.2    COMPARISON WITH DPPO

#### C.2.1    SETUP

DPPO (Ren et al., 2024), a very recent work, successfully fine-tunes diffusion policies using PPO, achieving state-of-the-art performance. Key tricks include fine-tuning only the last few denoising steps and fine-tuning DDIM sampling. Given that **this project was released around three weeks before the ICLR deadline**, we lacked sufficient time to fully adapt it to our tasks. Nevertheless, we conducted preliminary experiments comparing our approach with DPPO **on their tasks**. Even if DPPO is carefully tuned on their tasks, we are still able to beat it.

Specifically, we applied Policy Decorator (our approach) to the **two most challenging robotic manipulation tasks in their paper**: Square and Transport. We used the Diffusion Policy checkpoints provided by the DPPO paper as our base policies.

### C.2.2 RESULTS

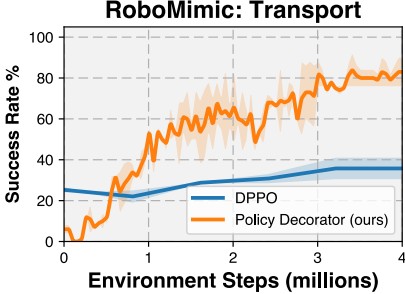

Figure 14: Policy Decorator (ours) vs. DPPO on the Transport task.

As shown in Fig. 14, our approach significantly outperforms DPPO on the Transport task. According to Figure 5 in the DPPO paper, DPPO requires approximately 16 million steps to converge to 80%+ success rate on the Transport task. In contrast, our Policy Decorator achieves this performance in only 4 million steps, demonstrating a **nearly 4x improvement in sample efficiency**.

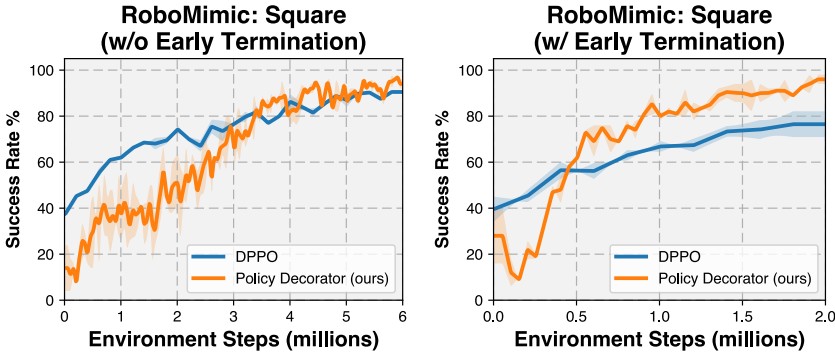

Figure 15: Policy Decorator (ours) vs. DPPO on the Square task.

On the Square task, our approach performs comparably to DPPO (left subfigure in Fig. 15). Upon further investigation, we discovered that DPPO uses a fixed episode length without early termination upon success signals. Empirically, this setup may negatively impact the sample efficiency of RL algorithms, as transitions after task completion contribute minimally to learning. Consequently, we conducted an additional experiment implementing early task termination upon success signals. **The results (right subfigure in Fig. 15) demonstrate that our approach outperforms DPPO in this more reasonable setup.**

### C.2.3 SUMMARY

**These experiments demonstrate that our method outperforms DPPO on challenging robotic manipulation tasks.** It is crucial to note that our approach is model-agnostic, whereas DPPO is restricted to a specific case of Diffusion Policy (where all predicted actions are executed in the environment, which is not the typical implementation of Diffusion Policy).

## D ADDITIONAL ABLATION STUDIES

This section includes additional ablation studies results about base policies, low-performing checkpoints, and PPO. In detail, Section D.1 discusses Policy Decorator also works with other types of

base policies (e.g., MLP, RNN, and CNN); Section D.2 demonstrates that Policy Decorator stays effective in improving low-performing BeT checkpoints; Section D.3 indicates that Policy Decorator is compatible with PPO as backbone RL algorithm.

### D.1 ADDITIONAL BASE POLICIES

To demonstrate that Policy Decorator is truly versatile to all types of base policy, we further experiment with model architecture of low representation power like MLP, BC-RNN, and CNN as well as low performance checkpoints of Behavior Transformer.

Fig. 16 demonstrates that the Policy Decorator significantly enhances the performance of MLP, BC-RNN, and CNN policies by interacting with environments.

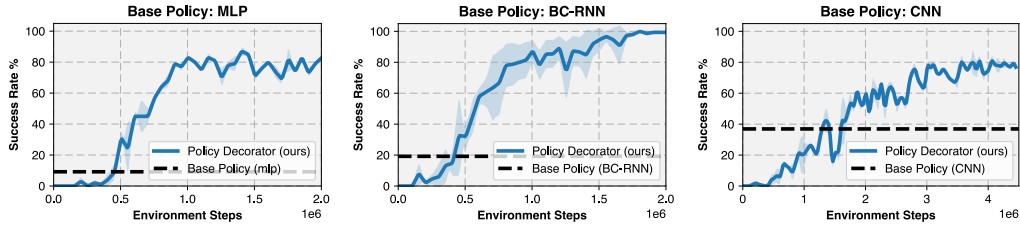

Figure 16: Policy Decorator with more base policies (MLP, BC-RNN, CNN) on TurnFaucet task through online interactions.

### D.2 USING OTHER CHECKPOINTS OF BASE POLICIES

As we claim that Policy Decorator is model-agnostic and is versatile to all types of base policies, it is necessary to demonstrate that it not only improves well-trained base policy but also improves low-performing checkpoints of base policy. Fig. 17 shows that the Policy Decorator achieves a substantial improvement in the low-performance BeT checkpoint.

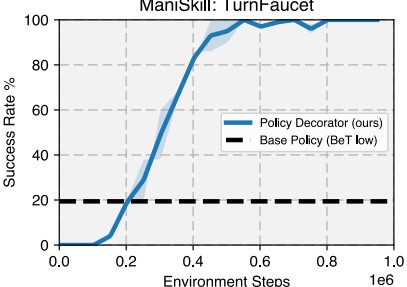

Figure 17: Policy Decorator with a low-performance BeT checkpoint.

### D.3 CHANGE BACKBONE RL ALGORITHM TO PPO

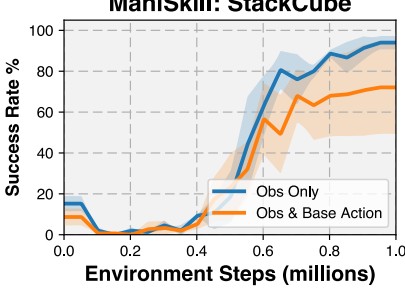

Figure 18: Use PPO as the backbone RL algorithm in our method, RL fine-tuning, and Residual RL.

While we use SAC as the backbone RL algorithm in our experiments due to its high sample efficiency, it is essential to demonstrate that the Policy Decorator can be integrated with other categories of RL algorithms, such as policy optimization, to provide greater flexibility. We changed backbone RL algorithm of our method, RL fine-tuning baseline, and residual RL baseline from SAC to PPO (Schulman et al., 2017). As shown

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

1. Q-function using an MLP

2. Q-function using a shared GPT backbone with the actor

3. Q-function using a separate GPT backbone

As shown in Fig. 24, we experimented with all the aforementioned Q-function architectures in SAC and PPO fine-tuning experiments. The results indicate that SAC fine-tuning with an MLP Q-function slightly improves the base policy, whereas SAC fine-tuning with the other two Q-function architectures does not yield such improvements. In contrast, PPO fine-tuning across all Q-function architectures demonstrates poor performance. Based on these observations, we chose to use the MLP Q-function in our fine-tuning baselines.

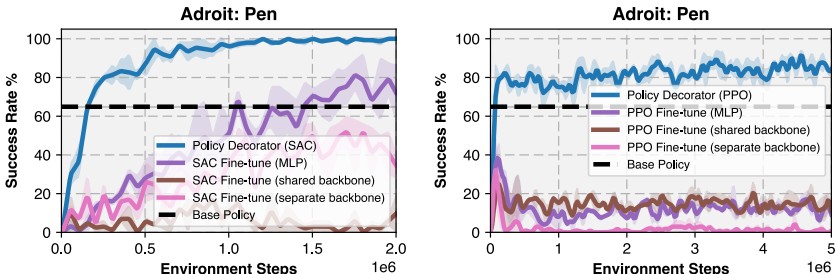

Figure 24: SAC/PPO fine-tuning with different critic architectures.

### F.5.2 EFFECT OF WARM-START IN Q FUNCTION TRAINING

Warm-starting Q function training is a widely used technique to ensure that the actor is updated with a reliable Q function. We also tried this technique in designing fine-tuning baselines. We experimented with a warm-start critic for a number of steps without training the actor. However, as shown in Fig. 25, this approach causes alpha, the learnable entropy coefficient in SAC, to increase massively, leading to an explosion in Q loss. We also compared vanilla fine-tuning with fine-tuning using a warm-start and fixed alpha. As indicated in Fig. 26, empirical results demonstrate that vanilla fine-tuning outperforms fine-tuning with a warm-start and fixed alpha. Upon closer examination, we found that fine-tuning with a warm-start and fixed alpha results in very unstable critic training. Therefore, we do not warm-start Q function training in our fine-tuning baselines.

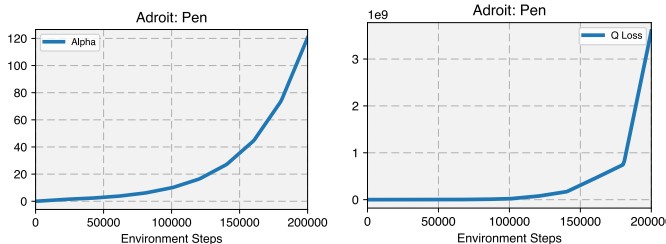

Figure 25: Critic warm start results in alpha and Q loss explosion when auto entropy tuning is enabled.

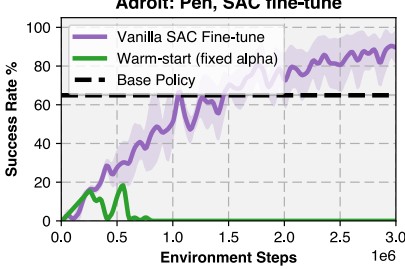

Figure 26: Warm-start the critic during fine-tuning.

## G  FAILURE OF NON-FINE-TUNING BASELINES

In this section, we analyze the poor performance of non-fine-tuning baselines in our experiments. We discusses the failure of vanilla Residual RL in Section G.1. We provides the explanations of failure of FISH in Section G.2.

### G.1  FAILURE OF VANILLA RESIDUAL RL

The residual RL baseline uses identical settings to our method, excluding the controlled exploration module. The primary failure mode of residual RL stems from 2 points:

1. Random residual actions in early training stages, causing the agent to deviate significantly from the base policy. This deviation leads to not getting any success signals for guiding learning. (see this video for an example).

2. Residual policy does not know it aims to minor fix the base policy, so during training, the average size of residual actions go beyond the average size of base policy actions, destroying the performance of base policy.

This is also supported by our ablation study (Fig. 10 and 11). As we gradually remove controlled exploration strategies (reducing H to 0 or increasing alpha to 1), our method approaches vanilla residual RL, resulting in deteriorating performance.

### G.2  FAILURE OF FISH

The primary failure mode of FISH stems from the extremely poor performance of non-parametric VINN policy in our experiments. See Section G.2.1 for the performance of VINN policy.

### G.2.1  VINN PERFORMANCE

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

. Given that **this project was released around three weeks before the ICLR deadline**, we lacked sufficient time to fully adapt it to our tasks. Nevertheless, we conducted preliminary experiments comparing our approach with DPPO *on their tasks*. Results indicate that our method significantly outperforms DPPO on their tasks. See Appendix C.2 for more details.