# OpenReview forum: "Policy Decorator: Model-Agnostic Online Refinement for Large Policy Model"
_ICLR.cc/2025/Conference — ICLR 2025 Poster_

### Official Review · Reviewer_mdBH · 2024-11-01

**Soundness:** 3
**Presentation:** 3
**Contribution:** 3
**Rating:** 8
**Confidence:** 3

**Summary:**

This paper presents Policy Decorator, a novel method that can fine-tune large, offline-trained imitation policies using Reinforcement Learning. Notably, the fine-tuning happens using a small network as a residual action generation policy on top of the pre-trained imitation learning policy. Policy Decorator works independently of the underlying imitation learned policy and simply wraps it in the fine-tuning loop.

**Strengths:**

This paper addresses an important research question and might lead to major enhancements in the research field. Current big models perform favorably but need costly demonstration data to work on novel but related tasks.
The well-written paper motivates the proposed method and clearly indicates the contributions.
Importantly, the method is simple but seems to be effective.

**Weaknesses:**

I don't have major concerns regarding the paper, but I believe the paper lacks a couple of related works that need to be discussed in my opinion:

- Jia et al. Towards diverse behaviors:  A benchmark for imitation learning with human demonstrations (ICLR 2024)
- Information maximizing curriculum: A curriculum-based approach for learning versatile skills (NeurIPS 2023)
- Goal-conditioned imitation learning using score-based diffusion policies (RSS 2023)

- Multi-modality might need more discussion (see Questions)

**Questions:**

I have some concerns regarding the effect of the residual policy on possible multi-modalities in the base distribution. The proposed base policies can represent multi-modal distributions (a mode does not refer to different input modalities in this case, but rather different modes in the underlying distribution given a state), making them a powerful method given that human data tends to be multi-modal. However, to my understanding, the residual policy is a Gaussian policy such that a mode collapse will happen once the fine-tuning starts. Does this mean that the overall fine-tuned policy is indeed maximizing performance, but therefore ignores other modes in the behavior?

This question could probably be answered by benchmarking on one of the proposed environments and data sets in the work by Jia et al. (see Weaknesses) that exactly analyzes the capabilities of learning multiple modes of imitation learning methods by providing task-specific diversity metrics.
Alternatively, plotting the policy for a multi-modal state (most often multi-modality occurs in the initial state) for the pre-trained and fine-tuned policy might provide some insights.

---

> ### Author Response · Authors · 2024-11-20
> **Author Response**
>
> We deeply appreciate your review and positive feedback! We are pleased that you find **our method simple but effective, and our paper well-written**. We value your efforts in helping us enhance the quality of our work. Below, we address your concerns.
>
> ---
>
> > lacks a couple of related works that need to be discussed
>
> Thank you for recommending these related works! **We have cited them in the related works section of the updated PDF.** We also discuss them below:
>
> D3IL [1] is a recent benchmark designed to quantitatively evaluate a model’s ability to learn multi-modal behavior. It offers a wide range of multi-modal tasks and demonstrations.
>
> IMC [2] and BESO [3] are both great imitation learning methods that handle multi-modal demonstrations. IMC [2] employs a mixture of experts policy, where each mixture component selects its own subset of the training data for learning. BESO [3] utilizes a generative, score-based diffusion model to learn general-purpose goal-specified policies from large uncurated datasets without rewards.
>
> *References:*
>
> [1] Jia, Xiaogang, et al. "Towards Diverse Behaviors: A Benchmark for Imitation Learning with Human Demonstrations." The Twelfth International Conference on Learning Representations.
>
> [2] Blessing, Denis, et al. "Information maximizing curriculum: A curriculum-based approach for learning versatile skills." Advances in Neural Information Processing Systems 36 (2024).
>
> [3] Reuss, Moritz, et al. "Goal-conditioned imitation learning using score-based diffusion policies." arXiv preprint arXiv:2304.02532 (2023).
>
> ---
>
> >  However, to my understanding, the residual policy is a Gaussian policy such that a mode collapse will happen once the fine-tuning starts. Does this mean that the overall fine-tuned policy is indeed maximizing performance, but therefore ignores other modes in the behavior?
>
>
>
> Adding the residual policy to the multi-modal base policy does not necessarily lead to mode collapse. Instead, we observed that the multi-modal property is typically preserved in our setups due to the residual policy's small scale. We illustrate this point through both an illustrative example and a real case study from our experiments.
>
>
>
>
>
> **Illustrative Example:**
>
> As demonstrated in  [Figure 101](https://sites.google.com/view/policy-decorator/home/iclr-rebuttal#h.6099lj1bl2ty), when a bimodal distribution (blue) is combined with a Gaussian distribution (orange), the sum distribution (green) still preserves its bimodal nature. This process effectively shifts the multi-modal distribution and adjusts the standard deviation of its modes. The multi-modal property is maintained as long as the Gaussian distribution's variance remains relatively small compared to the separation between modes.
>
>
> Implementation Notes:
> - The probability density function (PDF) of the bimodal distribution (blue): $f_{Bimodal}(x) = w_1 \cdot \mathcal{N}(x; \mu_1, \sigma_1^2) + w_2 \cdot \mathcal{N}(x; \mu_2, \sigma_2^2)$, where $\mathcal{N}$ represents Gaussian distribution.
> - The PDF of the Gaussian distribution (orange): $f_{Gaussian}(x) = \mathcal{N}(x; \mu_3, \sigma_3^2)$
> - The PDF of the sum of the two distributions (green) can be computed analytically: $f_{sum}(x) = w_1 \cdot \mathcal{N}(x; \mu_4, \sigma_4^2) + w_2 \cdot \mathcal{N}(x; \mu_5, \sigma_5^2)$, where $\mu_4=\mu_1+\mu_3, \sigma_4=\sqrt{\sigma_1^2+\sigma_3^2}, \mu_5=\mu_2+\mu_3, \sigma_5=\sqrt{\sigma_4^2+\sigma_3^2}$
> - The parameters used in the plot: $w_1=0.5, w_2=0.5, \mu_1=0.5, \mu_2=0.5, \mu_3=3, \sigma_1=1, \sigma_2=1, \sigma_3=1$
>
>
>
>
>
>
> **Real Case Study from Our Experiments:**
>
>
> To demonstrate the preservation of multi-modality in practice, we visualize action distributions from a specific state in the ManiSkill StackCube task, using Behavior Transformer as the base policy. We sampled 1000 actions from both base and residual policies, then applied PCA dimensionality reduction for visualization purposes. We use histograms to visualize these action samples. The results are shown in [Figure 102](https://sites.google.com/view/policy-decorator/home/iclr-rebuttal#h.ep0cmol9fju9).
>
> We can see that the base policy exhibits a clear bimodal distribution. When combined with the residual policy, the sum distribution maintains its bimodal nature while exhibiting slight shifts in position and variance. Note that the residual policy here is actually a squashed Gaussian distribution (as per SAC [4]) rather than a pure Gaussian, due to SAC's action bounds requirement. This practical example aligns well with our illustrative example, confirming that multi-modal property is preserved in our actual experiments.
>
>
> We have also included these results in Appendix L of the updated PDF.
>
> *References:*
>
> [4] Haarnoja, Tuomas, et al. "Soft actor-critic algorithms and applications." arXiv preprint arXiv:1812.05905 (2018).
>
>
> ---
>
> Hope the above clarifications and responses address your concerns. Thank you again for your time and effort!

---

> > ### Comment · Reviewer_mdBH · 2024-11-26
> >
> > I appreciate the author's efforts to clarify my concerns and questions. I believe this is a good paper and therefore keep my score.

---

### Official Review · Reviewer_UsJo · 2024-11-04

**Soundness:** 3
**Presentation:** 3
**Contribution:** 2
**Rating:** 5
**Confidence:** 3

**Summary:**

This paper considers how to use a model-agnostic residual policy to augment imitation learning policy in an offline-to-online transfer setting. The approach is very simple: freeze the base policy, and learn only a small residual policy that outputs a residual action which can be added to the base action. Online RL updates is only applied to the small residual policy. This approach has been shown to be highly effective across a variety of policies (like autoregressive Transformer and diffusion policies), and a variety of online RL algorithms (like SAC and variants of PPO).

**Strengths:**

- The approach is simple to implement, and the presentation is clear. The paper is easy to understand.
- The experimental evaluation is thorough.
- The performance seems very strong.

**Weaknesses:**

In my view, the most important weakness of this paper is this: The proposed approach is way too similar to the old idea of "Residual Policy Learning" [1, 2]. In fact, I would argue that **it is essentially the same as the old residual policy learning**. In the past, residual policy learning used a fixed controller. In this paper, residual policy learning uses a frozen pretrained policy.

As such, I don't think it is appropriate for the paper to rebrand this approach as "Policy Decorator", since it is basically the same as residual policy learning.

It appears that the essential finding can be summarized as: residual policy learning is more effective than finetuning for offline-to-online RL. I think saying that this finding is a new framework is confusing and misleading.

References:

[1] "Residual Policy Learning", Silver et al, 2018. arXiv:1812.06298

[2] "Residual Reinforcement Learning for Robot Control", Johannink et al, 2018. arXiv:1812.03201

**Questions:**

See the weakness section.

---

> ### Author Response · Authors · 2024-11-20
> **Author Response**
>
> Thank you so much for offering thoughtful and constructive feedback! We are delighted to know that you find **our experimental evaluation thorough, the performance strong, and the presentation clear**. Below, we address the concerns you raised.
>
> ---
>
> > I would argue that it is essentially the same as the old residual policy learning.
>
> We respectfully contest the viewpoint that “our method is essentially the same as the old residual policy learning.” **In contrast, we identify the uncontrolled exploration issue of vanilla residual RL (see example [here](https://sites.google.com/view/policy-decorator/home/random-residual-actions)), and propose a set of controlled exploration strategies to address it.** This distinction is discussed throughout the paper, including Sec. 1 (lines 91-101), Sec. 2 (lines 146-148), Sec. 4.2, Sec. 5.3 (lines 427-431), Sec. 5.4.1, Sec. 5.4.2, and Appendix G.1.
>
> More specifically, vanilla residual policy learning is hindered by **uncontrolled** random exploration in the early stages of training, which results in a lack of success signals to guide learning. To address this limitation, we introduce two **controlled exploration** strategies: bounded residual action and progressive exploration schedule.
>
> Our empirical results demonstrate that **these controlled exploration strategies are crucial for performance and learning efficiency**. As illustrated in Figures 6, 7, and 8, our approach significantly outperforms vanilla residual RL, particularly when the base policy's performance is not strong enough. The importance of our controlled exploration strategies is further validated through ablation studies presented in Figures 9, 10, and 11.
>
> ---
>
> > I don't think it is appropriate for the paper to rebrand this approach as "Policy Decorator".
>
> As explained above, our framework is distinct from residual RL, and we did acknowledge that our framework is built upon residual RL.
>
> The reason why we name our framework “Policy Decorator” is to highlight its model-agnostic nature. While this property may be less significant when using hand-crafted controllers as base policies (e.g., [1][2] in the reviewer's reference), **it becomes particularly crucial for improving large policy models**. This model-agnostic property is important here because: 1) improving large policy models is an increasingly relevant problem that demands attention; 2) the complicated architectural design of modern large policy models can make the fine-tuning process non-trivial. Therefore, we chose this name to highlight to robot learning researchers that residual-based learning can offer greater versatility and flexibility compared to naive fine-tuning when working with large policy models.
>
> ---
>
>
> Hope the above clarifications and responses address your concerns. If not, feel free to let us know, and **we are more than happy to answer additional questions**. If you feel that our rebuttal has addressed your concerns, we would be grateful if you would consider **revising your score in response**. Thanks so much for your time!

---

> > ### Comment · Reviewer_UsJo · 2024-11-25
> > **Reviewer Response**
> >
> > I thank the Authors for the response.
> >
> > I understand that the paper added "controlled exploration strategies", but in my view this part is very incremental as well. Action squashing was already applied by SAC by default, and here the paper is manually tuning and selecting an action bound for each individual environment. Having a schedule for epsilon-greedy is also very incremental; this seems like a design choice rather than a method or a framework in itself.
> >
> > I will maintain my score.

---

### Official Review · Reviewer_PZbK · 2024-11-04

**Soundness:** 3
**Presentation:** 4
**Contribution:** 3
**Rating:** 8
**Confidence:** 4

**Summary:**

The paper considers the setting of robot learning from demonstration with large policy models. The performance of existing supervised learning-based (i.e. behavioral cloning) approaches with large policy models is dependent on the quantity, quality, and diversity of the demonstrations, which in the robot setting are often resource-intensive to collect. To address these limitations, the paper considers fine-tuning the large policy models with reinforcement learning (with sparse rewards), and the paper identifies two major challenges: 1) non-differentiable components of the large policy models, and 2) performing many gradient updates on the policy model’s large number of parameters - as typically required during sparse-RL training - would be computationally expensive. The paper proposes learning a residual policy that can correct the behavior of the base policy. The paper proposes a method for controlling the exploration of the residual policy by progressively adding the residual policy’s output to that of the base policy throughout training. The paper conducts experiments on ManiSkill and AdroitHand tasks.

**Strengths:**

The paper presents a simple algorithm for fine-tuning large policy models in robot learning. The paper does a thorough job of validating baselines with ablation studies, providing justification for the design choices. The thorough empirical results - including the proposed method (Policy Decorator) showing improvement over the baselines across a number of tasks - is what establishes the quality of the paper and its contribution. In terms of clarity, the assumptions are clearly stated and the paper is written clearly.

**Weaknesses:**

One of the primary reasons to use large policy models in robot learning is they can model multimodal behavior, which a Gaussian policies parameterized by small neural networks cannot. In the paper, it remains unjustified whether adding the Gaussian residual policy to the base policy still retains the multimodal capabilities of the large policy model. (The answer may seem intuitive, but is worth justifying, as this is one of the principal motivations for using large policy models.)

The paper identifies fine-tuning large base policy models with RL as “prohibitively costly” due to the many gradient updates required and the large number of parameters. By using the large base policy (most of which are trained offline through a BC approach), this will dramatically improve the sample efficiency of RL. But, the point still stands that the large number of parameters may make gradient updates costly. However, no wall-clock times are reported, though they are mentioned when discussing the RLPD baseline (Appendix B.5) and Cal-QL baseline (Appendix F.3). Including wall-clock times could help inform how much more costly gradient updates on large policy models are (required by the baselines) than inference (required by both baselines and Policy Decorator).

The paper does not compare to standard, non-large policy model baselines, such as GAIL with an MLP or CNN policy. The paper specifically focuses on large policy models. However, it still remains an important baseline to observe whether large policy models are even necessary on these tasks.

The paper’s Policy Decorator approach, by controlling exploration through the Progressive Exploration Schedule, effectively allows the residual policy’s critic to be warm started before the residual policy has a noticeable effect on the behavior policy. It is therefore crucial to investigate whether warm starting the critic on the baselines improves their performance. The paper  addresses this in Appendix F. However, in Section F.5.2, the paper discusses warm starting in Q function training, noting that warm starting the critic causes a blowup in the entropy coefficient. Is it possible to fix the entropy coefficient during critic pretraining, but then unfreeze it during fine-tuning?

Some of the tasks in this paper have a dense reward that are easy to specify. This fact should be noted, as it would significant affect RL training by ameliorating the exploration problem. The paper is justified in wishing to consider the sparse-reward setting. For full clarity, however, the paper should identify which tasks may have access to an easily-specified dense reward.

**Minor Comments:**
- Define “diversity” of demonstration data (Line 038), as it is not discussed in the rest of the paper. (This can connect to the multimodal behavior discussed above.)
- There is a comma missing on Line 013, between “quantity” and “quality.”
- The number of seeds is not listed for Figure 7. (I presumed it followed the same as Figure 6.)
- In Figure 4’s caption, “schedule” is spelled wrong.

**Questions:**

See Weakness section.

---

> ### Author Response · Authors · 2024-11-20
> **Author Response (1/3)**
>
> We are so grateful for such a detailed review and positive evaluation! It is encouraging to receive recognition for **our thorough empirical results and clear writing**! We sincerely appreciate your efforts to help us improve the quality of our work. Below, we address the concerns you raised.
>
> *Note: The outcomes of additional experiments and the related discussions have been added to Appendix I, J, K, L, M of the updated PDF file.*
>
> ---
>
> > In the paper, it remains unjustified whether adding the Gaussian residual policy to the base policy still retains the multimodal capabilities of the large policy model.
>
>
> In this paper, applying a small residual action to correct a multi-modal base policy typically maintains its multi-modal property. We illustrate this point through both an illustrative example and a real case study from our experiments.
>
>
>
>
> **Illustrative Example:**
>
> As demonstrated in [Figure 101](https://sites.google.com/view/policy-decorator/home/iclr-rebuttal?#h.6099lj1bl2ty), when a bimodal distribution (blue) is combined with a Gaussian distribution (orange), the sum distribution (green) still preserves its bimodal nature. This process effectively shifts the multi-modal distribution and adjusts the standard deviation of its modes. The multi-modal property is maintained as long as the Gaussian distribution's variance remains relatively small compared to the separation between modes.
>
>
> Implementation Notes:
> - The probability density function (PDF) of the bimodal distribution (blue): $f_{Bimodal}(x) = w_1 \cdot \mathcal{N}(x; \mu_1, \sigma_1^2) + w_2 \cdot \mathcal{N}(x; \mu_2, \sigma_2^2)$, where $\mathcal{N}$ represents Gaussian distribution.
> - The PDF of the Gaussian distribution (orange): $f_{Gaussian}(x) = \mathcal{N}(x; \mu_3, \sigma_3^2)$
> - The PDF of the sum of the two distributions (green) can be computed analytically: $f_{sum}(x) = w_1 \cdot \mathcal{N}(x; \mu_4, \sigma_4^2) + w_2 \cdot \mathcal{N}(x; \mu_5, \sigma_5^2)$, where $\mu_4=\mu_1+\mu_3, \sigma_4=\sqrt{\sigma_1^2+\sigma_3^2}, \mu_5=\mu_2+\mu_3, \sigma_5=\sqrt{\sigma_4^2+\sigma_3^2}$
> - The parameters used in the plot: $w_1=0.5, w_2=0.5, \mu_1=0.5, \mu_2=0.5, \mu_3=3, \sigma_1=1, \sigma_2=1, \sigma_3=1$
>
>
>
>
>
>
> **Real Case Study from Our Experiments:**
>
> To demonstrate the preservation of multi-modality in practice, we visualize action distributions from a specific state in the ManiSkill StackCube task, using Behavior Transformer as the base policy. We sampled 1000 actions from both base and residual policies, then applied PCA dimensionality reduction for visualization purposes. We use histograms to visualize these action samples. The results are shown in [Figure 102](https://sites.google.com/view/policy-decorator/home/iclr-rebuttal#h.ep0cmol9fju9).
>
> We can see that the base policy exhibits a clear bimodal distribution. When combined with the residual policy, the sum distribution maintains its bimodal nature while exhibiting slight shifts in position and variance. Note that the residual policy here is actually a squashed Gaussian distribution (as per SAC [1]) rather than a pure Gaussian, due to SAC's action bounds requirement. This practical example aligns well with our illustrative example, confirming that multi-modal property is preserved in our actual experiments.
>
>
> We have also included these results in Appendix L of the updated PDF.
>
> *References:*
>
> [1] Haarnoja, Tuomas, et al. "Soft actor-critic algorithms and applications." arXiv preprint arXiv:1812.05905 (2018).
>
> ---
>
> > … how much more costly gradient updates on large policy models are (required by the baselines) than inference (required by both baselines and Policy Decorator).
>
>
>
> We conducted benchmarks on the Behavior Transformer's backward pass (gradient update) and forward pass (inference) running times. Results are shown in [Figure 103](https://sites.google.com/view/policy-decorator/home/iclr-rebuttal#h.ppfgvs7ewign)
>
> The results demonstrate that BeT's forward pass is significantly faster than its backward pass, with this gap becoming more pronounced as model size increases. This confirms that the backward pass constitutes the major training time bottleneck.
>
> **Implementation Details:**
> - Batch Size: 1024
> - GPU: NVIDIA GeForce RTX 2080 Ti
> - Results averaged over 100 independent runs
>
> Additionally, we present the actual training wall-clock time comparison below, demonstrating our Policy Decorator is indeed more time-efficient compared to naive fine-tuning:
>
>
> |                         | StackCube, BeT, 5M Env Steps |
> |:-----------------------:|:----------------------------:|
> | Policy Decorator (ours) |            7h 23m            |
> |     SAC Fine-tuning     |            33h 52m           |
>
>
>
> We have also included these results in Appendix J of the updated PDF.

---

> ### Author Response · Authors · 2024-11-20
> **Author Response (2/3)**
>
> > The paper does not compare to standard, non-large policy model baselines, such as GAIL with an MLP or CNN policy. … it still remains an important baseline to observe whether large policy models are even necessary on these tasks.
>
> Thanks for your suggestions! While our initial submission did not include the exact GAIL + MLP/CNN baseline, we had included ROT (an improved version of GAIL) as a baseline and conducted experiments with MLP and CNN (detailed in Appendix D.1).
> Following your suggestion, we have now implemented and tested the GAIL + MLP baseline. The results in [Figure 104](https://sites.google.com/view/policy-decorator/home/iclr-rebuttal#h.1jzcux79a3d3) show that this baseline achieves 0% success rate on StackCube and about 20% success rate on TurnFaucet after 3M environment interactions. These results are expected given that the demonstrations were collected task and motion planning (for StackCube) and model predictive control (for TurnFaucet) - resulting in naturally multi-modal distributions.
> Additionally, in offline imitation learning scenarios, MLP also performs significantly worse than large policy models, as shown in the table below:
> |                      | StackCube | TurnFaucet |
> |----------------------|:---------:|:----------:|
> | MLP                  |     0%    |     10%    |
> | Behavior Transformer |    71%    |     41%    |
> | Diffusion Policy     |    99%    |     55%    |
>
> These results together demonstrate that simple MLPs are insufficient for capturing multi-modal distributions and highlight the need for large policy models to effectively utilize multi-modal demonstrations.
>
> We have also included these results in Appendix K of the updated PDF.
>
> ---
>
> > Is it possible to fix the entropy coefficient during critic pretraining, but then unfreeze it during fine-tuning?
>
> Following the reviewer's suggestion, we fixed the entropy coefficient during the warm-start phase and enabled auto-tuning during subsequent fine-tuning. Results are shown in [Figure 105](https://sites.google.com/view/policy-decorator/home/iclr-rebuttal#h.gt8yyrge7q3b). From six independent runs, three of them still blow up upon unfreezing while the other three remained stable. This result indicates that this unfreezing strategy does not effectively address the training stability issue associated with warm-starting.
>
> We have also included these results in Appendix M of the updated PDF.
>
> ---
>
> > the paper should identify which tasks may have access to an easily-specified dense reward.
>
> Although we use sparse rewards in our experiments, all tasks in this paper do have existing human-engineered dense rewards:
>
> - [ManiSkill StackCube](https://github.com/haosulab/ManiSkill/blob/v0.5.3/mani_skill2/envs/pick_and_place/stack_cube.py#L134-L220):87 lines of code, 14 tunable hyperparameters
> - [ManiSkill PegInsertionSide](https://github.com/haosulab/ManiSkill/blob/v0.5.3/mani_skill2/envs/assembly/peg_insertion_side.py#L183-L264): 82 lines of code, 18 tunable hyperparameters
> - [ManiSkill TurnFaucet](https://github.com/haosulab/ManiSkill/blob/v0.5.3/mani_skill2/envs/misc/turn_faucet.py#L350-L390): 41 lines of code, 6 tunable hyperparameters
> - [ManiSkill PushChiar](https://github.com/haosulab/ManiSkill/blob/v0.5.3/mani_skill2/envs/misc/turn_faucet.py#L350-L390): 69 lines of code, 18 tunable hyperparameters
> - [Adroit Door](https://github.com/Farama-Foundation/Gymnasium-Robotics/blob/main/gymnasium_robotics/envs/adroit_hand/adroit_door.py#L293-L311): 18 lines of code, 9 tunable hyperparameters
> - [Adroit Hammer](https://github.com/Farama-Foundation/Gymnasium-Robotics/blob/main/gymnasium_robotics/envs/adroit_hand/adroit_hammer.py#L307-L324): 18 lines of code, 10 tunable hyperparameters
> - [Adroit Pen](https://github.com/Farama-Foundation/Gymnasium-Robotics/blob/main/gymnasium_robotics/envs/adroit_hand/adroit_pen.py#L314-L324): 11 lines of code, 8 tunable hyperparameters
> - [Adroit Relocate](https://github.com/Farama-Foundation/Gymnasium-Robotics/blob/main/gymnasium_robotics/envs/adroit_hand/adroit_relocate.py#L297-L313): 17 lines of code, 9 tunable hyperparameters
>
> As shown in the above codes, these human-engineered dense rewards are not as “easily-specified” as people may expect. They typically require dozens of lines of Python code and numerous tunable parameters. Designing these rewards manually involves extensive iteration over potential reward terms and tuning hyperparameters through trial and error. This process is laborious but critical for the success of human-engineered rewards.
>
> We have included these dense reward examples in Appendix I to illustrate the challenges in obtaining dense rewards and highlight why sparse-reward settings are preferable.

---

> > ### Author Response · Authors · 2024-11-20
> > **Author Response (3/3)**
> >
> > > Minor Comments
> >
> > - Diversity: This refers to the coverage of different scenarios, including scenes, object geometries, object textures, lighting conditions, etc.
> > - Missing comma: Fixed.
> > - Number of seeds in Figure 7: Yes, all experimental setups in Figure 7 are the same as Figure 6, including the number of seeds.
> > - Misspelling of “schedule”: Fixed.
> >
> > We appreciate your attention to detail in identifying these errors!
> >
> > ---
> >
> > Hope the above clarifications and responses address your concerns. Thank you again for your time and effort!

---

> > > ### Comment · Reviewer_PZbK · 2024-11-25
> > > **Reviewer response**
> > >
> > > Thank you for your thorough response. You have addressed all of my questions/concerns appropriately.

---

### Meta-Review · Area_Chair_2tZT · 2024-12-25

**Metareview:**

The introduction of the Policy Decorator presents a novel method for refining large policy models, which could enhance the performance of offline-trained models.
The performance improvements may not translate well to unseen tasks or environments. It would be better to include real-world applications, which could provide practical insights into the usability and effectiveness of the proposed method in real-world scenarios.

**Additional Comments On Reviewer Discussion:**

The reviewers tend to accept in general after the discussion phase.

---

### Decision · Program_Chairs · 2025-01-22

Accept (Poster)